# Temperament and sexual behaviour in the Furrowed Wood Turtle *Rhinoclemmys areolata*

**Francesca Maura Cassola[1], Yann Henaut[1]\*, José Rogelio Cedeño-Vázquez[1], Fausto Roberto Méndez-de la Cruz[2], Benjamín Morales-Vela[1]**

**1** El Colegio de la Frontera Sur, Unidad Chetumal, Chetumal, Quintana Roo, México, **2** Laboratorio de Herpetología, Departamento de Zoología, Instituto de Biología, Universidad Nacional Autónoma de México, Mexico City, México

\* yhenaut@ecosur.mx

**Data Availability Statement:** All relevant data are within the manuscript and its Supporting Information files.

## Abstract

The variation in temperament among animals has consequences for evolution and ecology. One of the primary effects of consistent behavioral differences is on reproduction. In chelonians some authors have focused on the study of temperament using different methods. In our research our first aim was i) establish a methodology to determine the degree of boldness among individuals *Rhinoclemmys areolata*. Our second aim was to ii) determine the role boldness plays during reproduction, with emphasis on courtship and copulation, considering a) the interactions between males and females, and b) competition between males. We used 16 sexually mature individuals of each sex. Males were observed in four different situations and 17 behavioral traits were recorded. We selected 12 traits that allowed us distinguish between the bolder and the shier individuals and found that five behavioral traits were specific for bolder individuals and five others for shier individuals. In a second step, we observed a male in presence of a female and recorded courtship behaviors and breeding attempts. Bolder individuals did not display courtship behaviors and just attempted to copulate. Shier individuals displayed courtship behaviors and copulation attempts were rarely observed. Finally, in the simulations that compared two males in the presence of a female we noticed that bolder individuals displayed courtship behaviors while the shier ones simply ignored the female. Our results first allowed us to determine which methodology is the best to determine temperament in turtles. Secondly, temperament seems to be an important factor in modulating interaction between males and females. Bolder individuals have an advantage during competition and display courtship behaviours only if other males are present. Shier males displayed courtship behaviors and only try to copulate when no competitors were present. These two different temperament-dependant strategies are discussed in terms of ecology, evolution and management.

## Introduction

Inter-individual behavioral differences in an animal species are consistent or largely maintained over time and in diverse situations [1] and sometimes between one population and the other [2]. These differences are termed "temperament", "behavioral syndrome" or "reactivity"

**Funding:** CFM and CVJR. El Colegio de la Frontera del Sur and CONACYT-OEA-AMEXCID scholarship program for the funds needed to cover expenses for data collection and specimen maintenance.

**Competing interests:** The authors have declared that no competing interests exist.

[1,3,4] by scientists. They have been attributed to the combined influences of genetic, social and environmental factors that influence the behavior of an individual [5], varying in correlation with the physiology of the animal [6,7] leading to mediate the response of an individual in different ecological situations [8].

Animal behavioral differences can have important implications for fitness [9,10] and have been shown to influence ecological processes such as dispersal, invasion, response to climate change and risk of extinction [11–13]. There is a high probability that temperament has a significant influence on the way in which animals respond to new environments (for example, new exhibitions or retention areas), to familiar and unfamiliar conspecifics, to strangers of other species, and to changes in their environment, affecting social compatibility, stability and success of a group [14]. Animal behavior and temperament can influence breeding capacity by directly connecting to a successful match (coupling success) or the number and viability of offspring produced (reproductive success) by the individuals [15–17]. Since temperament affects the way in which individuals react to difficult situations, it can be predicted how a male can compete, select and be compatible with a sexual partner. Therefore, variations in individual behavior are a source of information in the field of sexual selection and reproductive success [9,18].

Individuals may systematically differ in one or more temperament traits, such as boldness, aggressiveness, reactivity, sociability, exploration or activity levels [1]. A feature that has been extensively studied in the field of temperament is boldness, defined as the inclination of individuals to explore their environment and take risks in a new situation [19]. Boldness and shyness are considered as a continuum [20,21], at one extreme we find shier individuals and on the opposite side the bolder ones.

This spectrum appears to have an adaptive explanation in terms of costs and benefits, related to the type of response to a risk. In general, bolder individuals tend to be more active, feed more often and in risky areas as they are more likely to explore and move away from safe and well-known locations [22]. This temperament, however, leads to an increased risk of encountering predators [23] and exposing themselves to parasites [24], decreasing their survival rate. On the other hand, shy individuals engage in an opposite strategy giving primary importance to survival over reproductive productivity. This implies that this temperament tries to achieve a balance between survival and other needs of the species [21].

There are few studies addressing the relationship between success of couples and behavioral compatibility. Knowledge on these individual differences provides essential information for captivity management and animal welfare [14,21]. For example, it allows us to understand why some individuals have reproductive problems in captivity and why they fail to satisfy their reproductive potential, even if they are physically and physiologically in health [25]. A few years ago individuals suitable for mating were chosen based on their hypothetical genetic compatibility [26,27], their state of health [28] and / or stages of life history [29,30]. However, couples who represent a good genetic combination do not necessarily produce offspring, and behavioral incompatibility is often cited for this failure [31–34]. One way in which animal temperament can influence mating probability is through the effects associated with behavioral compatibility—the way individuals interact with each other [15]. When animals are paired for reproduction, some temperaments may be compatible, while others may be in conflict [14,35,36]. There are, for example, cases where behavioral similarity is the key to reproductive success, as in the case of zebra finches (*Taeniopygia guttata*) and cockatiels (*Nymphicus hollandicus*) [37,38]; while, in other species, different or even opposite temperaments can attain high reproductive performances as in the giant panda (*Ailuropoda melanoleuca)* and the black rhino (*Diceros bocornis)* [15,25,39].

Recently, boldness has been studied in aquatic turtles. The recognition between males and the reduction of competitive interactions are related to the boldness of males [40]. They showed that the response to chemical stimuli of known or unknown males of the Mediterranian Turtle *Mauremys leprosa*, depends on the level of boldness, in particular the shiest males are those who avoid encountering other males, of the same or different species contributing to the stabilization of social systems and reducing the frequency and intensity of aggressive encounters between males. Different methods are used to evaluate temperament in turtles. Reactivity level is measured using the straightening response, which is the time it takes for an individual to become upright after being turned upside down on its carapace [41]. A similar method has been used to study anti-predator responses in the European pond turtle *Emys orbicularis* [42]. Another method consists in placing individuals in a new environment: here, the latency to move from an initial position in an arena was used to evaluate exploration in the red-eared slider turtle *Trachemys scripta* [43] and in the Eastern hermann *Eurotestudo boettgeri* [44]. The presentation of a threatening stimuli was used to measure reactivity and the effect of new objects on investigative behaviors while researching exploration in the Agassiz´s desert tortoise (*Gopherus agassizii*) [4]. Boldness in the Eastern box turtle (*Terrapene carolina*) was evaluated by recording the time taken to emerge from the shell and to move after a short period of confinement [45]. Evidently, there are diverse methods applicable to turtles; however, none of the aforementioned studies tried to apply more than one method to a single species. Therefore, a unique methodology for the study of boldness in turtles is still lacking.

Unfortunately, there is very little research that specifically addresses how temperament may be related to the reproductive behavior of turtles, particularly its influence on male competition behavior, courtship and coupling. Thus, the aims of this study were to (i) determine the temperament of *Rhinoclemmys areolata* and provide a method that not only ascertains temperament for this species but also other species of terrestrial turtle (ii) understand if its temperament could be a factor in modulating the interaction between male and female, and (iii) between males during competition for a female.

## Materials and methods

### Ethical statement

The investigation was carried out in compliance with the institutional ethical standards and norms in force. All the animals come from private loans from the city of Chetumal (Quintana Roo–Mexico). Verbal consent was requested from the owners, and it was explained how the turtles would be used and for what objective. The individuals on loan were destined exclusively for specific research purposes, respecting all known needs for animal welfare, and were returned to their owners at the end of the data collection.

We were in possession of the permit N ˚ SGPA / DGVS / 002491/18 issued by Secretaria de Medio Ambiente y Recursos Naturales (SEMARNAT) for the collection of this species.

Experimental protocol was approved by Ethical Committee from the "El Colegio de la Frontera Sur", Mexico.

### Object of study

For the experiment, we used 16 males and 16 females of the furrowed wood turtle *Rhinoclemmys areolata*, all sexually mature. This species occurs in Yucatán Peninsula, Cozumel Island, Northern Guatemala, Belize, Northwestern Honduras, where it inhabits savannah, thorny scrub, broad-leaved forests, swampy and gouache areas [46]. It is used as a pet, for food and traditional medicine [47]. It is listed as Near-threatened by the IUCN Red List [48]. It is also known as "Chak pool" or "Mojina" [47]. It is characterized by 1 to 2 red, orange or yellow

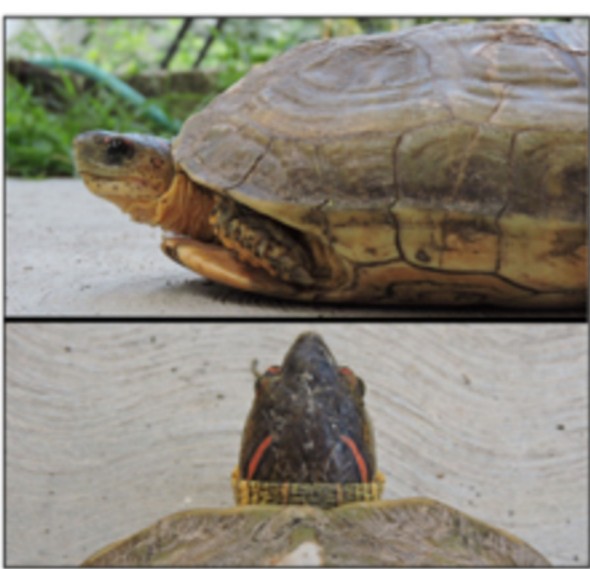
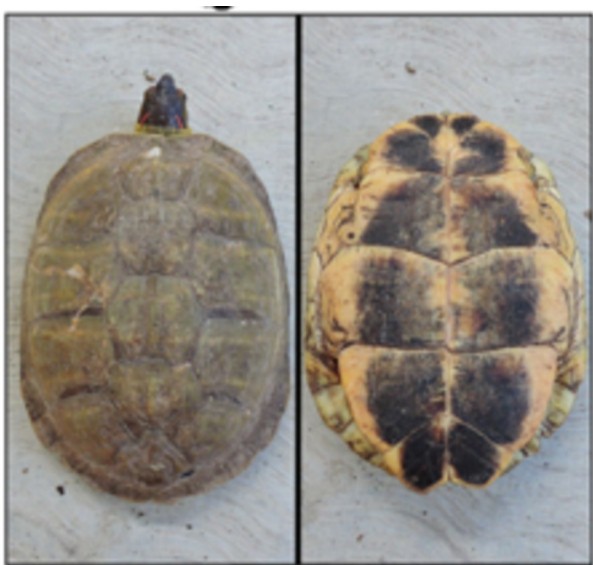

**Fig 1. Details of *Rhinoclemmys areolata* morphology.**

stripes on the top of the head, and the plastron is yellow with a dark spot of variable size (Fig 1). It presents a low level of sexual dimorphism, males having a longer and thinner tail than females, and also having a slightly more concave plastron; however, this is characteristic is not always detectable (Fig 2). It is a terrestrial species, whose only connection with the aquatic environment is during courtship and copulation, although this information solely derives from two anecdotic observations events where the individuals were kept in pools without the presence of a large terrestrial environment [49,50]. Courtship and copulation could be induced by spraying the individuals with water [46].

## Turtles' management and setup

Individuals of *Rhinoclemmys areolata* were kept within a 16 x 5 m enclosure, located in the facilities of El Colegio de la Frontera del Sur (ECOSUR) Unidad Chetumal (18˚ 32'37.4"N, 88˚ 15 ' 48.0"W). The area was fenced and divided into two parts of approximately the same size (5 x 8 m) containing male and female individuals, respectively. This division prevented any reproductive activity in the absence of the observer. It also ensured that the female did not store semen and therefore be less reactive to males. Females from some species of turtle are able to retain vital sperm for up to 4 years; however, it is not yet clear in which species, therefore we cannot dismiss that it could also occur in *R. areolata* [51].

The area inside the fence resembled as close as possible the natural habitat of *R. areolata*, by maintaining the presence of trees, foliage, natural soil and adding water sites (Fig 3).

Turtles were fed with fruit, vegetables and meat on alternate days. Inside the maintenance area, they were free to move, to find refuge and bathe in small pools. The water in the maintenance area was changed daily. They were subject to natural temperatures and rainfall for as long as they were inside the area in order to sustain optimum environmental conditions for their welfare.

All simulations were performed in a mobile enclosed area (2 x 2 m) (Fig 4). Each simulation was recorded through two cameras (YiLiteCam and Motorola Camera). The first video camera was positioned above the observation area while the second was moved manually by the

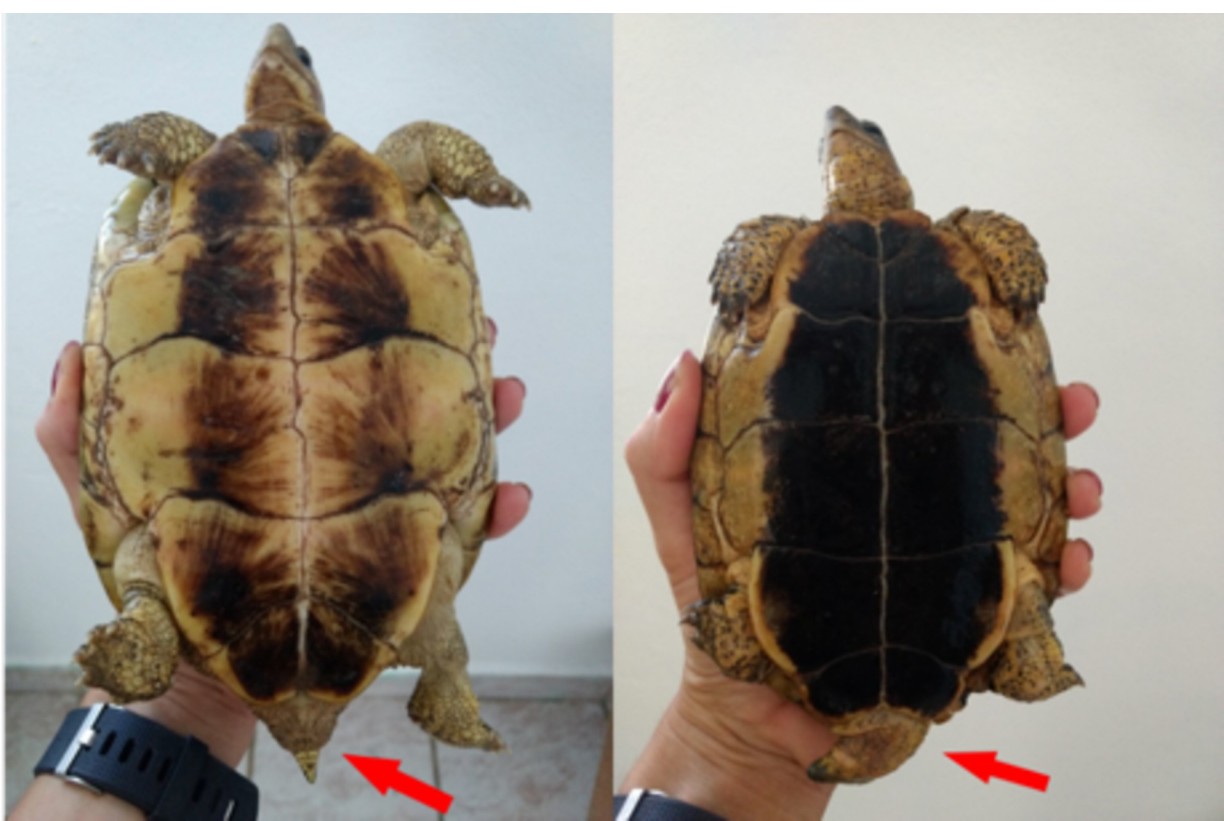

**Fig 2. Detail of female (left) and male (right).** Differences between the tails (red arrow) are visible; females have a shorter and thicker tail, while males have a longer and thinner one.

observer in the event that details of specific events were required. There were no food or small pools in the experimental area that could have had an effect on turtle behavior [41].

Each simulation took place in the open air, on sunny days and during the hours of maximum turtle activity (8.30am-11.30am and 05.30pm—06.30pm) when temperatures were not very high.

Animals were separated after each observation session to avoid any additional intersexual interactions in absence of the observer. The observation area was moved to an adjacent area after each simulation to reduce the time elapsed between experiments and the recovery time for environmental conditions; subsequently a new group of turtles was placed into the observation area in order to repeat the simulation.

## Experiments

**Determination of temperament of *Rhinoclemmys areolate*.** It was necessary to identify stable behavioral traits over time and in different situations. For this aim,16 males were observed individually and in isolation in four different situations:

- Reaction to manipulation: a male is removed from confinement and handled briefly for 1 minute.

- Reaction to risk of predation: a male is removed from confinement, handled briefly and gently released in an overturned position, that is, on his carapace.

**Fig 3. Captive living area of *Rhinoclemmys areolata* at the ECOSUR campus.**

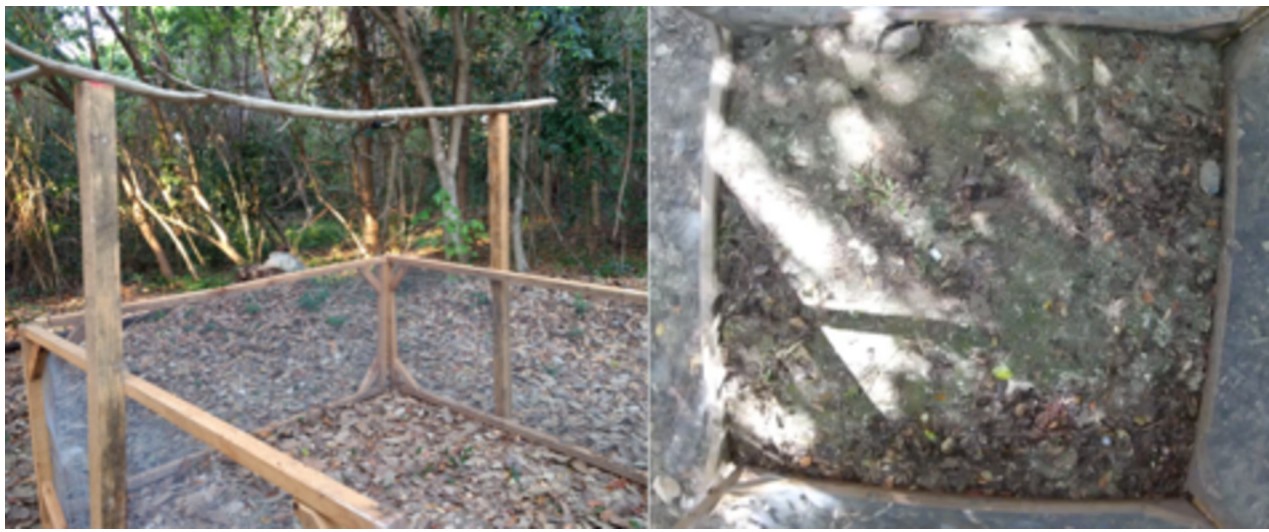

**Fig 4. Observation area viewed from the side and above.**

- Reaction to a new environment: a male is taken from the confinement and released into the experimental area unknown to the individual. (30min)

- Reaction to the introduction of a new object: a male is removed from confinement, released into the experimental area and, after 5 minutes, an unknown object is suddenly presented (stuffed dog-like animal). (30min)

In order to organize males according to their level of boldness-shyness we: (i) extrapolated behavioral events of each simulation from video recordings, (ii) converted them into variables, (iii) defined occurrence (= 1) and absence (= 0) of each event obtaining binomial values; (iv) defined frequency and (v) time in seconds when possible, obtaining quantitative values; (vi) for the latter, the moment an abrupt change of slope occurred within the data range was recorded, allowing the classification of males as shy or bold (S1 Table).

Each event was considered as a variable and inserted into an Excel sheet. A boldness index was given to each male, dividing the sum of "occurrences" of variables that could describe a bold temperament by the total number of variables. This analysis allowed us to classify individuals into groups, according to their degree of boldness.

Subsequently, variables were analysed using a correlation matrix and the obtained values were inserted into a multidimensional scaling (MDS) that enabled the creation of a map showing relative positions between different events and relative distances.

Finally, a Mann-Whitney test was applied to determine which variables were significant in determining temperament.

Before initiating interaction experiments, we tested the influence of turtle length (that is considered a good indication of the age of each individual) on temperament. We implemented this test to ensure that our results were not the consequence of turtle size or age. Both groups obtained were analysed in relation to straight carapace length (SCL) and straight plastron length (SPL) by applying a Mann-Whitney test (S2 Table).

**Temperament as a modulating factor during the interaction between male and female.** 16 groups of turtles consisting of one male and one female were created randomly. Each couple was released, sequentially, within the experimental area, each individual inside a metal box. They were soaked with water using an artificial sprayer to promote reproductive behavior and later freed to move around for 30 minutes.

For this experiment, the same individuals were not used more than once.

Each group remained sequentially in the experimental arena and was carefully observed.

As for the first simulations, we (i) extrapolated behavioral events from each simulation, (ii) converted them into variables, (iii) defined occurrence (= 1) and absence (= 0) of each event obtaining binomial values; (iv) defined frequency and (v) time in seconds when possible, obtaining quantitative values; (vi) for the latter, the moment an abrupt change of slope occurred within the data range was recorded (S3 and S4 Tables).

Each event was considered as a variable and inserted into an Excel sheet. Variables were analysed using a correlation matrix and the obtained values were inserted into a multidimensional scaling (MDS) allowing the creation of a map showing relative positions between different events and relative distances.

**Temperament as a modulating factor during the competitive interaction between males in the presence of a female.** Eight experimental groups were randomly created, each consisting of two males and one female. For this experiment, the same individuals were only used once.

The individuals were placed equidistant within the experimental area, soaked with water using an artificial sprayer to promote reproductive behavior and released to move around for 30 minutes.

Each group remained in the experimental area and was carefully observed.

As for the first simulations, we (i) extrapolated behavioral events from each simulation, (ii) converted them into variables, (iii) defined occurrence (= 1) and absence (= 0) of each event obtaining binomial values; (iv) defined frequency and (v) time in seconds when possible, obtaining quantitative values; (vi) for the latter, the moment an abrupt change of slope occurred within the data range was recorded (S5 and S6 Tables).

Each event was considered as a variable and inserted into an Excel sheet. Variables were analysed using a correlation matrix and the obtained values were inserted into a multidimensional scaling (MDS) that allowed the creation of a map showing relative positions between different events and relative distances.

## Results

### Determination of temperament of *Rhinoclemmys areolata*

Of 17 behavioral traits observed (S1 Table), we selected 12 for study (Table 1). For each male, the presence or absence of each of these traits was defined (Table 2). For this study, we used "bolder" and "shier" terms instead of "bold" and "shy" since boldness is a continuous spectrum and it is not possible to create sharp and fixed fractures for group definition.

In order to create two groups for the following experiments, we considered all turtles with an index less than or equal to 0.5 as shier, while those with an index greater than 0.5 bolder. As a result, individuals were divided into 9 bolder and 7 shier males (Table 2).

**Table 1. Variables selected for simulations, subdivision used in correlation analysis and multidimensional scaling with relative abbreviation.**

| Variable | Subdivision for analysis | Abbreviation |
|---|---|---|
| Kicking during manipulation (KM) | Fast kicking | FK |
| | Slow kicking | SK |
| Neck stretched during manipulation (NeSM) | Yes neck stretched manipulation | YNeSM |
| | No neck stretched manipulation | NoNeSM |
| Hiding inside shell during manipulation (HSM) | Yes hide shell manipulation | YHSM |
| | No hide shell manipulation | NoHSM |
| Neck retracted during manipulation (NeRM) | Yes neck retracted manipulation | YNeRM |
| | No neck retracted manipulation | NoNeRM |
| Straightening time during predation (STP) | High straightening time predation | HSTPr |
| | Low straightening time predation | LSTPr |
| Hiding inside shell during predation (HSP) | Yes hide shell predation | YHSPr |
| | No hide shell predation | NoHSPr |
| Walking time in a new environment (WTNE) | High walking time new environment | HWTNE |
| | Low walking time new environment | LWTNE |
| Hiding in a new environment (HNE) | Yes hide new environment | YHNE |
| | No hide new environment | NoHNE |
| Quiet time in new environment (QTNE) | High quiet time new environment | HQTNE |
| | Low quiet time new environment | LQTNE |
| Crossing a new environment (CrNE) | Yes cross area | YCrNE |
| | No cross area | NoCrNE |
| Exploration of stuffed toy (ExST) | Yes exploring stuffed toy | YExST |
| | No exploring stuffed toy | NoExST |
| Exploring time stuffed toy (ExST) | High time exploring stuff toy | HExST |
| | Low time exploring stuffed toy | LExST |

Table 2. Presence (= 1) or absence (= 0) of variables selected for simulation and boldness index values.

| Turtle Name | Sum variables | Boldness Index | KM | NeSM | HSM | NeRM | STD | HSPr | WTNE | HNE | QTNE | CrNE | ExST | ExTST |
|---|---|---|---|---|---|---|---|---|---|---|---|---|---|---|
| Alfredo | 9 | 0,75 | 1 | 1 | 1 | 1 | 1 | 0 | 0 | 1 | 0 | 1 | 1 | 1 |
| Benedicto | 12 | 1 | 1 | 1 | 1 | 1 | 1 | 1 | 1 | 1 | 1 | 1 | 1 | 1 |
| Carlos | 8 | 0,67 | 1 | 1 | 1 | 1 | 1 | 0 | 0 | 1 | 1 | 1 | 0 | 0 |
| Hector | 8 | 0,67 | 1 | 1 | 1 | 0 | 1 | 0 | 1 | 1 | 1 | 0 | 1 | 0 |
| Nestor | 8 | 0,67 | 1 | 1 | 1 | 1 | 0 | 0 | 1 | 1 | 1 | 1 | 0 | 0 |
| Erik | 7 | 0,58 | 0 | 0 | 0 | 0 | 1 | 1 | 1 | 1 | 1 | 1 | 1 | 0 |
| Garry | 7 | 0,58 | 0 | 1 | 0 | 0 | 1 | 0 | 1 | 1 | 1 | 0 | 1 | 1 |
| John | 10 | 0,83 | 0 | 1 | 1 | 1 | 1 | 1 | 1 | 1 | 1 | 1 | 1 | 0 |
| Oliver | 8 | 0,67 | 1 | 0 | 1 | 0 | 1 | 0 | 1 | 1 | 1 | 1 | 1 | 0 |
| Francesco | 5 | 0,41 | 1 | 1 | 1 | 0 | 0 | 0 | 0 | 1 | 1 | 0 | 0 | 0 |
| Kevin | 3 | 0,25 | 0 | 0 | 1 | 0 | 0 | 0 | 0 | 0 | 0 | 0 | 1 | 1 |
| Marc | 5 | 0,41 | 1 | 1 | 1 | 1 | 0 | 0 | 0 | 1 | 0 | 0 | 0 | 0 |
| Patricio | 3 | 0,25 | 0 | 0 | 1 | 0 | 0 | 0 | 0 | 0 | 0 | 0 | 1 | 1 |
| Denis | 6 | 0,5 | 1 | 1 | 1 | 0 | 1 | 0 | 0 | 1 | 0 | 1 | 0 | 0 |
| Ian | 6 | 0,5 | 1 | 1 | 1 | 1 | 1 | 1 | 0 | 0 | 0 | 0 | 0 | 0 |
| Lorenzo | 5 | 0,41 | 1 | 0 | 1 | 0 | 1 | 1 | 0 | 0 | 0 | 0 | 1 | 0 |

In grey background color the individuals considered bolder, in white those considered shier.

Straight carapace length and straight plastron length did not show a significant relationship with temperament (Carapace: Mann-Whitney U test: U = 20, P = 0.25; Plastron: Mann-Whitney U test: U = 19, P = 0.21) (S7 Table).

Rearranging the variables according to presence, frequency and times (Table 1), and through the multidimensional scaling, we obtained a total of 10 variables: five correlated with the shiest individuals and five associated with the boldest individuals (Fig 5).

These variables were characteristic of the shier turtles: (i) hiding in a new environment, (ii) not crossing a new environment, (iii) a high duration of quiet time in a new environment, (iv) high straightening time during predation, and (v) low walking time in a new environment. Variables for boldest turtles were: (i) not hiding in a new environment, (ii) cross a new environment, (iii) low quiet time in a new environment, (iv) low straightening time during predation, and (v) high walking time in a new environment (Fig 5). Four of those variables were also statistically significant for the discrimination of the two groups (Fig 6, S8 Table).

## Temperament as a modulating factor during the interaction between male and female

As results for the relation between temperaments and male and female interaction, we observed behaviors linked with courtship and others with mating.

Regarding courtship, we determined 16 behaviors (S3 Table), but we only used 11 for the analysis, excluding behaviors showed by most of the individuals and those that did not present any differences between groups.

We determined if the behaviors occurred or not and how often to obtain their frequency (low or high) (Table 3). We observed that the following traits corresponded to shier individuals (Fig 7): (i) pushing the female, (ii) blocking the female, (iii) frontal approach with stretched legs and retracted neck (iv) pushing the ground with front leg, (v) approaching when excited. On the other hand, some bolder turtles did not show courtship behavior while in others this was not very frequent. Bolder individuals displayed the following behaviours: (i) do not block

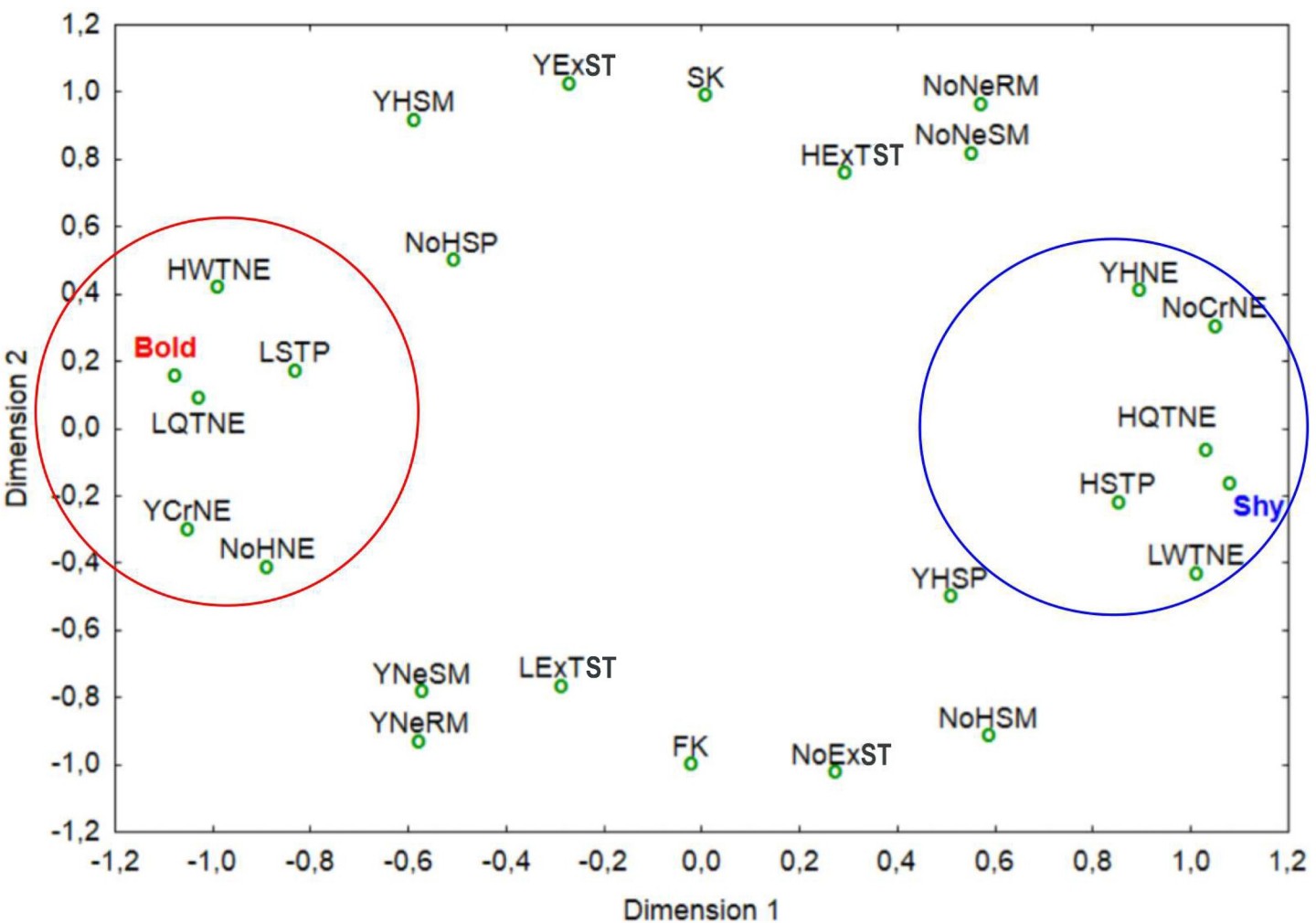

**Fig 5. Distribution of the variables (behaviors) of the individuals.** The red circle shows variables associated with the boldest individuals; the blue circle identifies behaviors associated with the shiest ones.

the female, (ii) do not push it, (iii) do not approach excitedly, (iv) do not walk backwards, (v) low frequency of frontal approach with stretched legs and extended neck (Fig 7).

Four behaviors (Table 4, S4 Table) were observed for mating. None of these behaviors was displayed by the shiest individuals, while the boldest were more associated with intent of mounts, and mount at high frequency (Fig 8, S1 Video).

## Temperament as a modulating factor during the competitive interaction between males in the presence of a female

We obtained 12 behaviors related to courtship (Table 5, S5 Table). We considered the temperament of the two males together with a female and observed that courting variables are concentrated around the boldest regardless the temperament of their opponent (Fig 9). In contrast, the shiest individuals did not display particular behaviors with the exception of blocking the female during tracking (Fig 9) (S2 Video).

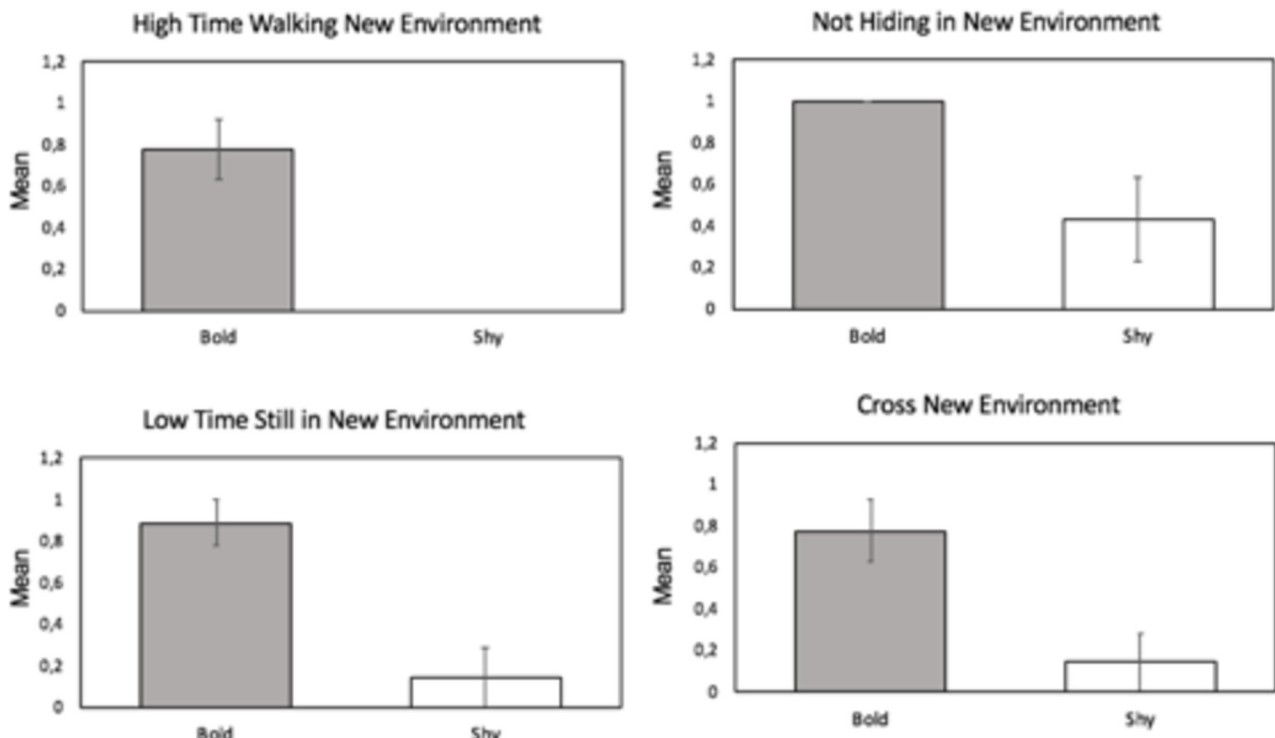

**Fig 6. Mean values and standard errors of the four significant binomial variables.** High time walking new environment, Bolder: mean±SE = 0,78 ±0,1, Shier: mean±SE = 0±0, Mann-Whitney U test: U = 7, p = 0.00*3; Not hiding new environment, Bolder: mean±SE = 1±0, Shier: mean±SE = 0,43 ±0,2, Mann-Whitney U test: U = 13,5, p = 0.013*; Low time quiet new environment, Bolder: mean±SE = 0,89±0,1, Shier: mean±SE = 0,14±0,14, Mann-Whitney U test: U = 37, p = 0.004*; Cross new environment, Bolder: mean±SE = 0,78±0,14, Shier: mean±SE = 0,14±0,1, Mann-Whitney U test: U = 37, p = 0.017*.

## Discussion

A variety of methods were used separately to determine temperament in aquatic and terrestrial turtles [4,41–45]. This is the first paper that tests all of these methods. In this study, we tested reaction to: manipulation, predation risk, a new environment and the introduction of a novel object with the aim of finding consistency, to determine accurately behaviors and in consequence the best method to define the level of boldness in turtles. Behaviors observed in a new environment appear to be the most reliable for determining boldness in turtles: hiding and limited walk for the shiest, quiet (shiest) or not quiet (boldest), walking frequently and crossing the new environment for the boldest. These results coincide with two previous works [43,44].

The test with the risk of predation also demonstrates that the time taken to straighten is associated with boldness, as observed in previous studies [41,42].

Our work focused only on males, and determining temperament in females or juveniles may be different even though the exploration of a new environment may be comparable irrespective of the sex and age of turtles that generally share the same life strategy. Therefore, to assure that behaviors associated with the reaction to a new environment can provide an improved method, we suggest further studies that examine this in females and juveniles.

An interesting element that emerges in this study is the modulation of males´ behavior.

When a bold male is alone with a female, he often skips courtship behaviors to devote himself to stimulating the female for mounting and mating; when the same male is found in the presence of another conspecific male, a potential sexual competitor, he first engages in courtship behaviors towards the female. A similar result has been observed in the common water

**Table 3. Courtship behaviors obtained from simulations male vs female, subdivision used in correlation analysis and multi-dimensional scale with relative abbreviation.**

| Variable | Subdivision |
|---|---|
| Frontal approach with stretched legs and stretched neck | High Frequency (HFASLSN) |
| | Low Frequency (LFASLSN) |
| | No Approach (NoFASLSN) |
| Frontal approach with stretched legs and retracted neck | High Frequency (HFASLRN) |
| | Low Frequency (LFASLRN) |
| | No Approach (NoFASLRN) |
| Cross snouts | High Frequency (HCrS) |
| | Low Frequency (LCrS) |
| | No Cross (NoCrS) |
| Push ground with front leg | High Frequency (HPGL) |
| | Low Frequency (LPGL) |
| | No Push (NoPGL) |
| Block female | Yes Blocking (YBlock) |
| | No Blocking (NoBlock) |
| Pushing female | Yes Pushing (YPush) |
| | No Pushing (NoPush) |
| Walk backwards | High Frequency (HWB) |
| | Low Frequency (LWB) |
| | No Walk (NoWB) |
| Excited male is approaching | Yes excited male (YEMA) |
| | No excited male (NoEMA) |
| Quiet male explores surroundings | Yes Explore (YQMExp) |
| | No Explore (NoQMExp) |
| Cloaca sniffing | High Frequency (HCS) |
| | Low Frequency (LCS) |
| Remote tracking | Yes Remote (YRT) |
| | No Remote (NoRT) |

strider *Aquarius remiges* (Gerridae, Insecta), where the most active and aggressive males spend most of their time searching for a female as well as presenting the highest copulation rates [52]. In the common lizard *Zootoca vivipara* (Lacertidae, Reptilia), males' behavior together with the predatory context influence the mating preference of the females. Those that have not been exposed to predatory signals prior to mating, mate more often with more active males. If, on the other hand, these have been subjected to predatory signals, they tend to choose less active males that are more likely to have a longer life expectancy and to avoid the more active males in order to increase the survival of their offspring in an environment with risk of predation [53]. In our simulations, mating attempts of shier males were very rare; when the shier turtle is in presence of a competitor and a female, it ignores the female. In summary, the boldest individuals court females, but only if another male is present; otherwise these males immediately mount and attempt to copulate. Shier males will only court if other males are absent; even then mating is rare. This kind of behavior in the presence of a competitor was observed in *Mauremys leprosa* where responses to chemicals cues depended on boldness; in particular, shy turtles avoided chemical cues from familiar and unfamiliar bolder males [41]. However, it is possible shier animals require more time to display courtship and mating behavior. Therefore, we can argue that the degree of boldness can influence males' reproductive behavior, in decision making and their reaction according to the situation. In terms of costs and benefits,

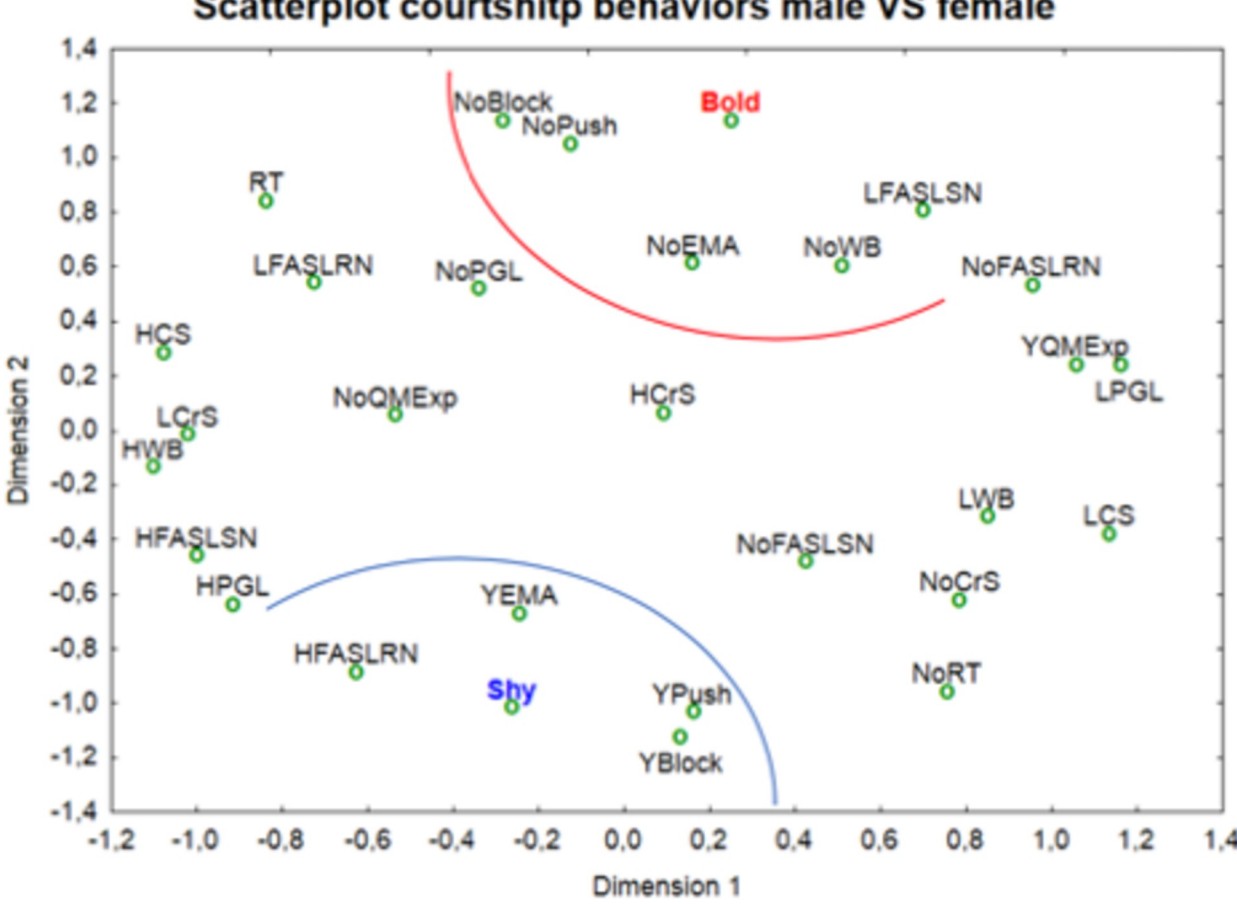

**Fig 7. Distribution of courtship variables (behaviors) of the male individuals.** The red curve includes variables associated with the boldest individuals; the blue curve congregates behaviors associated with the shiest individuals.

boldness can be considered a factor that allows individuals to evaluate how to act during courtship and mating considering the presence of conspecifics of both sexes. Shier turtles could avoid injuries and the possibility of losing energy when a bolder male is competing. In our

**Table 4. Mating behaviors obtained from simulations male vs female, subdivision used in correlation analysis and multi-dimensional scale with relative abbreviation.**

| Variable | Subdivision |
| --- | --- |
| Neck stimulation | High Frequency (HNS) |
| | Low Frequency LNS) |
| | No Stimulation (NoNS) |
| Mount attempt | High Frequency (HMA) |
| | Low Frequency (LMA) |
| | No Mount attempt (NoMA) |
| Coitus | Yes Coitus (YCo) |
| | No Coitus (NoCo) |
| Mount | High Frequency (HM) |
| | Low Frequency (LM) |
| | No Mount (NoM) |

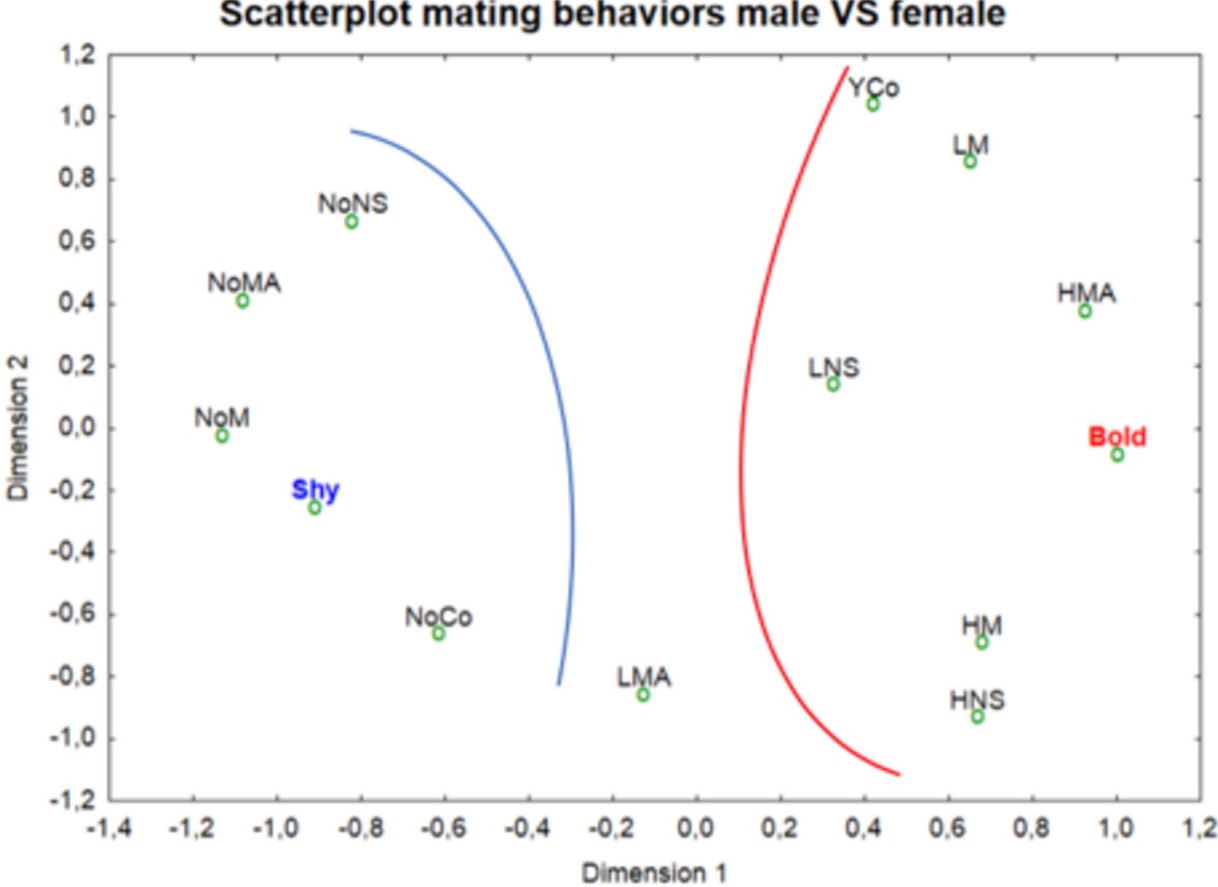

**Fig 8. Distribution of mating variables (behaviors) of the male individuals.** The red curve encompasses variables associated with the boldest individuals; the blue curve includes behaviors associated with the shiest individuals.

experimental contexts, we could not directly observe injury phenomena between males; however, in other experiments, we have observed that males have bitten each other when trying to

**Table 5. Courtship behaviors obtained from simulations male vs male vs female used in correlation analysis and multi-dimensional scale with relative abbreviation.**

| Variable | Abbreviation |
| --- | --- |
| Approaching cloaca | AC |
| Stretching neck towards female | SN |
| Cloaca sniffing | CS |
| Lateral sniffing | LS |
| Tracking | T |
| Quiet alongside female | Q |
| Bite | Bi |
| Blocking female | Bl |
| Lean leg to female | LL |
| Walk backwards | WB |
| Push female | PF |
| Push ground with leg | PGL |

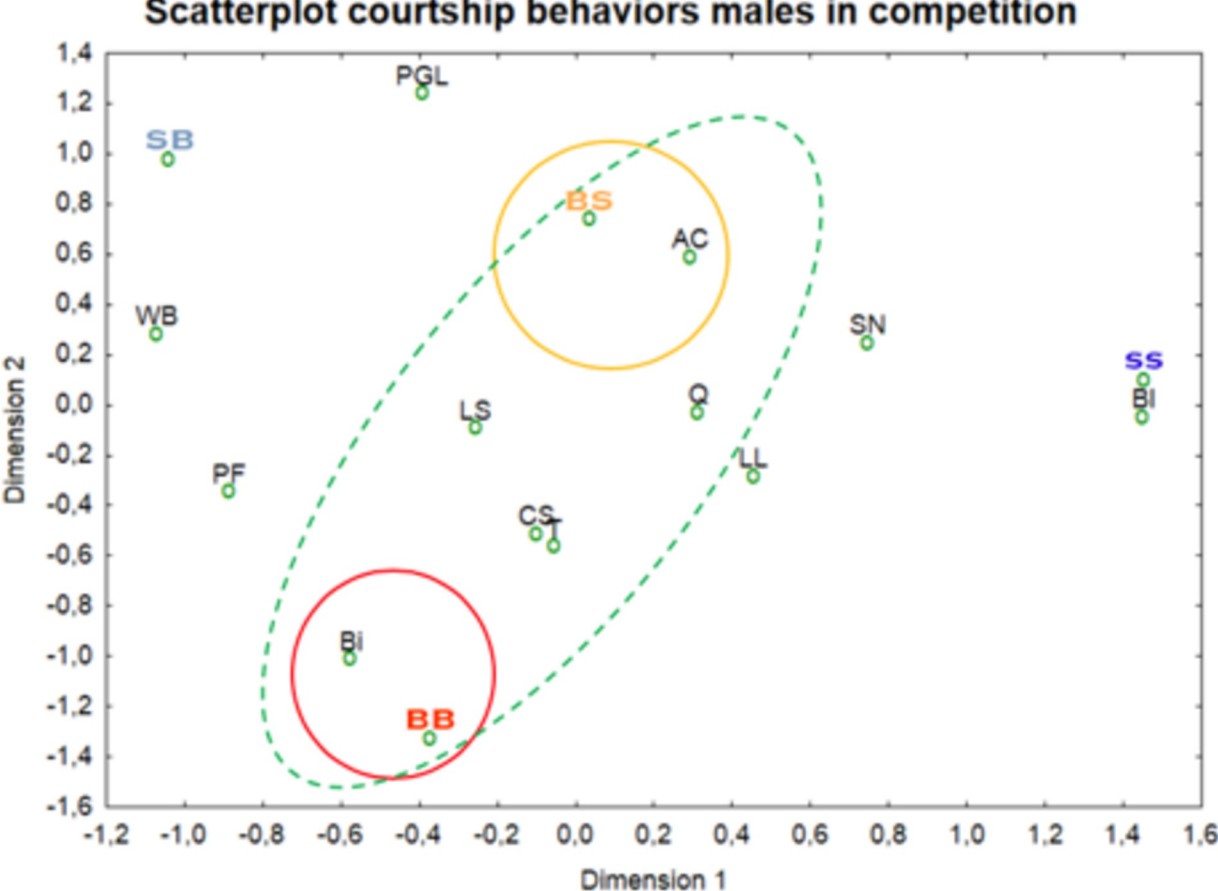

**Fig 9. Distribution of courtship variables (behaviors) of the individuals.** The red circle delimits variables that are highly associated with the boldest competing males; the yellow circle encompasses those highly associated with a bold male competing with a shy individual; the green ellipse includes behaviors associated with the boldest individual regardless of competitor.

approach the same female. In other species, male competition could be very costly because of fights, injuries, territory care and predatory danger during courtship. Some reptiles avoid or reduce their response to such competition and could provide an alternative strategy to prevent damage and predator costs, as shown in the Iberian wall lizard *Podarcis hispanica*, which uses chemical cues during competitive matches to recognize competitors and reduce fight intensity and resulting costs [54]. The success of different strategies dependent on temperament is to some extent subject to environmental constraints such as the presence of predators as observed for other animals [1]. For example, the broadhead skink *Eumeces (Plestiodon) laticeps* shows a sensitivity in predation risk during reproductive opportunities: when females were present, isolated and mate-guarding males initiate courtship, but isolated males permitted closer approach by a predator than did mate-guarding males. Furthermore, there is a high level of latency displayed by isolated males when a predator is close to the female, allowing them to balance predation risk with courtship behavior [55].

Researchers have already discussed that behavior plays a key role in animal survival and evolution, behavioral traits can help species to adapt to rapid changes in the environment [56], and survive to natural selection [57] as much as morphological and physiological traits. Differences in sexual behavior related to boldness or other behavioral traits can be an honest signal

of mate quality [58]. Therefore, identifying these behavioral traits that can affect population viability is important to provide helpful information to integrate conservation plans [59].

The information obtained in this study represents an important tool for the conservation of reptiles and in particular for land turtles. Worldwide, turtles represent one of the vertebrate groups with the highest risk of extinction [60]. In recent years, many researchers have ensured the importance of not only increasing the behavioral information of the species, but also implementing these data in their conservation and monitoring strategies [14,61].

Assumptions have been made in the past about the likely relationship between temperament and reproductive behaviors [62], but to our best knowledge, this is the first work that formalizes it experimentally for land turtles.

The achievements in this study can be used as a first step for compatibility plans for turtles, in order to combine temperament traits that may contribute to successful breeding couples [14].

Temperament studies in couple compatibility have significant implications in *ex situ* conservation programs, since specific traits (such as aggressiveness and fear) have been related to reproductive success in some species raised in zoos [15]. In addition, certain temperaments may be more likely to thrive in captive environments, artificial selection for or against specific temperament traits may accelerate domestication processes, making animals raised for conservation less suitable for release in wildlife [32].

Our research showed that bolder males need less time to mount and mate, with perhaps higher fitness than shier ones.

Another application of our findings is the possibility of conserving the whole range of boldness. Considering behavioral variation, it is important to preserve boldness since it allows conservationists to reintroduce animals with all behavioural traits, giving them proper tools to survive in novel environments [57]. Considering this, in order to not to lose the behavioral pool during captive reproduction, we must be aware of the needs of different individuals. Therefore, our study demonstrates that shy turtles probably need more time for mating and need to be alone with the female, otherwise they lose interest. For the boldest animals, the necessary times are lower and they are potentially able to reproduce even with the presence of conspecifics.

## Supporting information

**S1 Table. List of responses found in behavioral simulation (occurrence = 1, absence = 0), with relative abbreviation and subdivision applied for statistical analysis.** The first column shows the names of each individual, bolder individuals have a grey backgrounds while those considered shier have white.
(DOCX)

**S2 Table. Straight carapace and straight plastron length measurements of each male turtle.** Individuals considered bolder have grey backgrounds while those considered shier have white.
(DOCX)

**S3 Table. List of courtship behaviors during male and female simulation (occurrence = 1, absence = 0), with relative abbreviation and subdivision applied for statistical analysis.** The first column shows the names of each individual. Bolder individuals have grey backgrounds while shier individuals have white.
(DOCX)

**S4 Table. List of mating behaviors during male and female simulation (occurrence = 1, absence = 0), with relative abbreviation and subdivision applied for statistical analysis.** The

first column shows the names of each individual, bolder individuals have grey backgrounds while shier have white.
(DOCX)

**S5 Table. List of courtship behavior during male vs male competition in presence of female (occurrence = 1, absence = 0), with relative abbreviation and subdivision applied for statistical analysis.** The first column shows the names of each individual, bolder individuals have red backgrounds while shier have blue. Columns 2,3,4 and 5 indicate pairs by temperament (BB = boldVSbold, BS = boldVSshy, SS = shyVSshy, SB = shyVSbold).
(DOCX)

**S6 Table. List of mating behavior during male vs male competition in presence of female (occurrence = 1, absence = 0), with relative abbreviation and subdivision applied for statistical analysis.** In the first column, there are names of each individual, in grey background color the bolder and in white the shier. Columns 2,3,4 and 5 indicate pairs by temperament (BB = boldVSbold, BS = boldVSshy, SS = shyVSshy, SB = shyVSbold).
(DOCX)

**S7 Table. Statistical results of Mann-Whitney test on length measurements.**
(DOCX)

**S8 Table. Statistical results of Mann-Whitney test on variables selected for boldness index, significant variables are indicated by** *.
(DOCX)

**S1 Video. Shy courtship and bold mount.**
(MP4)

**S2 Video. Shy and bold male in presence of a female.** Shy male is walking around the arena while bold male courtships the female.
(M4V)

## Acknowledgments

We would like to thank the citizens of Chetumal who gave and loaned individuals of *R. areolata* for this research; José Alberto García Angulo for his help in the design and construction of the turtle maintenance and observation areas; Dr. Gabriele Gheza, Martina Anna Belli and Antonin Mareš for their help in grammatical and linguistic revision.

We would like to express special thanks to Dr David Crews, who has critically read our manuscript, providing useful and substantial comments that have allowed us to improve our work.

## Author Contributions

**Conceptualization:** Francesca Maura Cassola, Yann Henaut, José Rogelio Cedeño-Vázquez, Fausto Roberto Méndez-de la Cruz, Benjamín Morales-Vela.

**Data curation:** Francesca Maura Cassola.

**Formal analysis:** Francesca Maura Cassola, Yann Henaut.

**Funding acquisition:** José Rogelio Cedeño-Vázquez.

**Investigation:** Francesca Maura Cassola, Yann Henaut.

**Methodology:** Francesca Maura Cassola, Yann Henaut, José Rogelio Cedeño-Vázquez, Fausto Roberto Méndez-de la Cruz, Benjamín Morales-Vela.

**Project administration:** Yann Henaut.

**Supervision:** Yann Henaut.

**Writing – original draft:** Francesca Maura Cassola, Yann Henaut.

**Writing – review & editing:** Francesca Maura Cassola, Yann Henaut, José Rogelio Cedeño-Vázquez, Fausto Roberto Méndez-de la Cruz, Benjamín Morales-Vela.

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
