## [Decision Letter · Decision Letter 0]

29 Jul 2020

PONE-D-20-14068

Temperament and sexual behaviour in the Furrowed Wood Turtle Rhinoclemmys areolata

PLOS ONE

Dear Dr. Henaut,

Thank you for submitting your manuscript to PLOS ONE. After careful consideration, we feel that it has merit but does not fully meet PLOS ONE’s publication criteria as it currently stands. Therefore, we invite you to submit a revised version of the manuscript that addresses the points raised during the review process.

I wish to apologize up front as I was hoping to provide you two sets of reviews. However, this chelonian species is understudied, and thus, several investigators, who work with other turtle species, politely declined. However, I am pleased to report that a leading authority in the field has reviewed your manuscript. This individual expressed enormous enthusiasm for the model and findings. To further strengthen what already appears to be a solid study, this individual has kindly added corrections and suggestions to the PDF version. Please also see the other enclosed document provided. As one who is interested in turtles, I look forward to seeing the revised manuscript.

We look forward to receiving your revised manuscript.

Kind regards,

Cheryl S. Rosenfeld, DVM, PhD

Academic Editor

PLOS ONE

Journal Requirements:

2. In your Methods section, please provide additional details regarding the animals used in your study and ensure you have described the source. For more information regarding PLOS' policy on materials sharing and reporting, see https://journals.plos.org/plosone/s/materials-and-software-sharing#loc-sharing-materials.

3. In your Methods section, please provide additional details regarding participant consent from the owners of the animals. In the ethics statement in the Methods and online submission information, please ensure that you have specified (1) whether consent was informed and (2) what type you obtained (for instance, written or verbal). If the need for consent was waived by the ethics committee, please include this information.

4. In your Methods section, please provide additional information regarding the permits you obtained for the work. Please ensure you have included the full name of the authority that approved the collection site access and, if no permits were required, a brief statement explaining why.

5. In your Methods section, please include a comment about the state of the animals following this research. Were they returned, released or housed for use in further research?

6. In your Methods section, please provide additional location information of the collection sites, including geographic coordinates for the data set if available.

7. Please include a copy of Table 5 which you refer to in your text on page 17 (Table 3 2x).

Reviewers' comments:

Reviewer's Responses to Questions

**Comments to the Author**

1. Is the manuscript technically sound, and do the data support the conclusions?

Reviewer #1: Yes

2. Has the statistical analysis been performed appropriately and rigorously? 

Reviewer #1: Yes

3. Have the authors made all data underlying the findings in their manuscript fully available?

Reviewer #1: Yes

4. Is the manuscript presented in an intelligible fashion and written in standard English?

Reviewer #1: Yes

5. Review Comments to the Author

Reviewer #1: Review Comments to the Author

TURTLES ARE A VERY UNDERSTUDIED ANIMAL IN TERMS OF BEHAVIORAL ANALYSIS. THIS IS AN INNOVATIVE AND BROAD STUDY THAT SHOULD GENERATE CONSIDERABLE INTEREST.

Please see my attached review.

6. PLOS authors have the option to publish the peer review history of their article (what does this mean?). If published, this will include your full peer review and any attached files.

Reviewer #1: **Yes: **DAVID CREWS

---

## [Author Response · Author response to Decision Letter 0]

20 Aug 2020

A) Editor comments (changes in green in the document)

1) We revised Plos One style requirements.

2) To answer the editor comments about ethical statements (points 2, 3, 4, 5, 6), in the ethical declaration we added the following paragraph:

“All the animals come from private loans of the city of Chetumal (Quintana Roo – Mexico). Verbal consent was requested from the owners, and it was explained how the turtles would be used and for what objective. The individuals on loan were destined exclusively for specific research purposes, respecting all known needs for animal welfare, and were returned to their owners at the end of the data collection.

We were in possession of the permit N ° SGPA / DGVS / 002491/18 issued by Secretaria de Medio Ambiente y Recursos Naturales (SEMARNAT) for the collection of this species.”

3) Page 18 line 325: in Materials and Methods we corrected the mistake in the table caption, it was a typing error.

4) Pages 30-32 lines 581-620: We added the captions for supporting information files.

5) We revised all the references and updated all the new citations.

B) Responses to Reviewer (in yellow in the document):

Short Title

6) Page 1 Line 10: we deleted the short tittle after we revised Plos One style requirements (see Editor comments).

Summary

7) Page 2 Lines 29-31: According to the reviewer comment, we changed the phrase “Finally, we observed two males in presence of one female and noticed that bolder individuals then displayed courtship behaviors while the shier just ignore the female.” with “Finally, in the simulations that compared two males in the presence of a female we noticed that bolder individuals showed courtship behaviors while the more shy ones simply ignored the female.” The change was made because the result does not come from a single anecdote as the previous sentence suggested, but from multiple individuals and repetitions. 

8) Page 2 Lines 34-35: following the previous correction this sentence should now coincide with results from multiple experiments and not just one, as it previously seemed.

9) Page 2 line 48: we deleted the Key Word after we revised Plos One style requirements (see Editor comments).

Introduction

10) Page 3 line 44-45: as suggested by the reviewer we suppressed the term personality considered as anthropomorphic and controversial. Also, as suggested, we mentioned now the term “reactivity” and suppressed “permanently”.

11) Page 3 line 54: we changed “breed” with “breeding” as suggested by the reviewer.

12) Page 3 line 56: as suggested by the reviewer we added the citation “Gowaty PA, Drickamer LC, Schmid-Holmes S. Male house mice produce fewer offspring with lower viability and poorer performance when mated with females they do not prefer. Anim. Behav. 2003;65: 95-103”.

13) Page 3 line 59: as suggested by the reviewer we added the citation “Ah‐King, M., & Gowaty, P. A. (2016). A conceptual review of mate choice: stochastic demography, within‐sex phenotypic plasticity, and individual flexibility. Ecology and evolution, 6(14), 4607-4642”.

14) Page 4 line 69: as suggested by the reviewer, we added “giving primary importance to survival over reproductive productivity” to specify behaviors.

15) Page 4-5 lines 85-90: as suggested by the reviewer we suppressed the unnecessary sentence and add references (Beach FA and Leboeuf BJ 1967, Coital behaviour in dogs. I. Preferential mating in the bitch. Animal behaviour, 15(4) 546-548, Sánchez-Macouzet O, Rodríguez C, Drummond H. 2014 Better stay together: pair bond duration increases individual fitness independent of age-related variation. Proc. R. Soc. B 281: 20132843).

16) Page 4-5 lines 85-90: As suggested by the reviewer in the discussion, we added the following paragraph in the introduction “There are, for example, cases where behavioral similarity is the key to reproductive success, as in the case of zebra finches (Taeniopygia guttata) and cockatiels (Nymphicus hollandicus) [31-32] ; while, in other species, different or even opposite temperaments can reach high reproductive performances as in the giant panda (Ailuropoda melanoleuca) and in the black rhino (Diceros bocornis) [12, 25, 33].”

17) Page 5 line 91: we created a new sentence as suggested by the reviewer deleting “and the study shows that”

18) Page 5 line 98: as suggested by the reviewer we changed boldness for reactivity.

19) Page 5 line 105: as suggested by the reviewer we changed boldness for reactivity.

20) Pace 5 line 110: we changed “attempt to create” with “creating” as suggested by the reviewer.

Material and Methods

21) Page 6 line 125: we deleted the hyphenation in “North-western”.

22) Page 7 line 146: we change the tense of “avoid” to “avoided”.

23) Page 7 line 147: we were talking about sperm storage that was studied in different turtles’ species but there aren’t any available work about this species. For a better understanding we changed the word “recollecting” with “storing”.

24) Page 8 line 167: during the trial we used water only to spraying the individuals, no pool or drinking point were used, therefore there was no reason to refill something. The water we used to spray was always new and after every trial we changed the position of the observation area into a dry one.

25) Page 10 line 228: we changed the tense from “will not be” to “were”.

26) Page 10 line 232: We used “sequentially” to explain that the experiments were done one after the other, but the term is maybe not adequate. We deleted this word to avoid confusion.

Results

27) Page 12 line 262: we changed the word “organisms” with "turtles” as suggested by the reviewer.

28) Page 12 line 265: we changed “didn’t” with “did not” as suggested by the reviewer.

29) Page 12 line 273: as suggested by the reviewer we change “for the shier” by “characteristic of the shier turtles”.

30) Page 12 line 276: As suggested by the reviewer we changed “for the boldest” by “. Variables for the boldest turtles were”.

Discussion

31) Page 20 line 361: we changed the beginning of the sentence from “Also, our” to “Our” as suggested by the reviewer.

32) Page 21 line 386: we deleted “the”.

33) Page 21 line 386: we added “animals”.

34) Page 21 line 393-394: according to referee’s commentary, we add in the sentence that it was observed for other species “In other species, male competition could be very costly”.

35) Page 22 lines 407-408: in order to rewrite the sentence as suggested by the reviewer, we changed “respond” with “adapt” and we delete “pass under the clutches of” to add “survive to”.

36) Page 22 line 418: according to the reviewer comment we suppressed” behavioral trait” and added “temperament” to be more specific in our affirmation.

37) Page 22: we deleted the paragraph “In the study of Spoon et al (2006), it can be seen that compatible pairs (male-female) of the Cockatiel Nymphicus hollandicus showed higher levels of coordination in parental care and raised more chicks; another example is given by Carlstead et al (1999a, b) who observed in Black Rhino Diceros bocornis that smaller enclosures determined the breeding of dominant males, while domination decreased in males housed in larger enclosures. In addition, dominance levels were related to reproductive success, with the conclusion that the number of births per year was higher in pairs of rhinos composed of a submissive male and a dominant or aggressive female. This information gives an idea of the optimal conditions for the breeding of captive rhinos, including large enclosures, which facilitate submissive behavior in males, which in turn affects the compatibility of couples; in the giant panda Ailuropoda melanoleuca, Martin-Wintle et al (2017) conducted behavioral studies to further improve reproductive performance and found that specific combinations of temperament traits showed better reproductive performance than others. Males who were more aggressive than their partners were more likely to mate and produce puppies than when the female had a higher level of aggression than the male or even couples with less fearful males, regardless of the fear of the female, obtained better reproductive results. The application of these results to captive breeding management strategies should result in higher reproductive rates and the production of more candidates for the giant panda reintroduction program.” and added a resumed version in page 4 lines 95-100.

38) Page 22 lines 428: we deleted “focused on boldness degree of males”, same with “therefore they could have a” and added “with perhaps”.

39) Page 23 lines 435 and 437: we deleted “has allowed to understand” and added “demonstrates” and added “animals”.

40) Page 23: as suggested by the reviewer and to avoid repetition of the same idea, we deleted the sentence “By another way in term of evolution, being shiest may have some advantages and, if not for reproduction, they may be for antipredatory behaviour, making shiest individuals, that are more prudent in term of exploration, and as we observed in the new environment, generally less predated than boldest individuals as discussed for other species.”

41) Page 24: as suggested by the reviewer we deleted the last paragraph.

Bibliography

42) Following the suggestions of the reviewer we made changes in the citations. We added the following works:

- Gowaty PA, Drickamer LC, Schmid-Holmes S. Male house mice produce fewer offspring with lower viability and poorer performance when mated with females they do not prefer. Anim. Behav. 2003;65: 95-103.

- Ah‐King M, Gowaty PA. A conceptual review of mate choice: stochastic demography, within‐sex phenotypic plasticity, and individual flexibility. Ecol. Evol. 2016;6: 4607-4642.

- Beach FA and Leboeuf BJ Coital behaviour in dogs. I. Preferential mating in the bitch. Anim Behav. 1967; 15: 546-548.

- Sánchez-Macouzet O, Rodríguez C, Drummond H. Better stay together: pair bond duration increases individual fitness independent of age-related variation. Proc. R. Soc. B. 2014; 281: 20132843.

And deleted:

- Cable RN. Oh, Behave: When wildlife behavior matters in Conservation. Mich. J. Sustain. 2013;1

- Cooper Jr WE. Variation in escape behavior among individuals of the striped plateau lizard Sceloporus virgatus may reflect differences in boldness. J. Herpetol. 2009;43:495-502.

---

## [Decision Letter · Decision Letter 1]

9 Sep 2020

PONE-D-20-14068R1

Temperament and sexual behaviour in the Furrowed Wood Turtle Rhinoclemmys areolata

PLOS ONE

Dear Dr. Henaut,

Thank you for submitting your manuscript to PLOS ONE. After careful consideration, we feel that it has merit but does not fully meet PLOS ONE’s publication criteria as it currently stands. Therefore, we invite you to submit a revised version of the manuscript that addresses the points raised during the review process.

The original expert reviewer reviewed the revised version, and I have also gone through it. I fully agree with the reviewer that the revised manuscript did not take into consideration many of the detailed and helpful comments the reviewer provided on the original manuscript. The work is exciting. However, for it to be readable and accepted by the scientific community, manuscript presentation and clarity is important. I encourage you to go through the original reviewer's comment and additional ones provided this round and address each one of them. In so doing, it will greatly strengthen the manuscript. This individual has graciously spent considerable time reviewing both versions and suggesting changes. Please do not submit another version that has been superficially modified. Have others read it over beforehand for grammar, syntax, and overall clarity. If more time is needed in editing the work, please contact the journal.

We look forward to receiving your revised manuscript.

Kind regards,

Cheryl S. Rosenfeld, DVM, PhD

Academic Editor

PLOS ONE

Reviewers' comments:

Reviewer's Responses to Questions

**Comments to the Author**

1. If the authors have adequately addressed your comments raised in a previous round of review and you feel that this manuscript is now acceptable for publication, you may indicate that here to bypass the “Comments to the Author” section, enter your conflict of interest statement in the “Confidential to Editor” section, and submit your "Accept" recommendation.

Reviewer #1: (No Response)

2. Is the manuscript technically sound, and do the data support the conclusions?

Reviewer #1: Yes

3. Has the statistical analysis been performed appropriately and rigorously? 

Reviewer #1: Yes

4. Have the authors made all data underlying the findings in their manuscript fully available?

Reviewer #1: Yes

5. Is the manuscript presented in an intelligible fashion and written in standard English?

Reviewer #1: No

6. Review Comments to the Author

Reviewer #1: In the first review of this MS, I made a particular effort to instruct how the MS should be edited for format, syntax, grammar etc. While those remarks were incorporated, there was no effort to revise the overall MS. I have now re-written the abstract and parts of the paper. The authors need to have an accomplished English speaking author go through the MS to make this an contribution rather than a paper to be ignored because it is so poorly written.

7. PLOS authors have the option to publish the peer review history of their article (what does this mean?). If published, this will include your full peer review and any attached files.

Reviewer #1: **Yes: **David Crews

---

## [Author Response · Author response to Decision Letter 1]

23 Oct 2020

We responded editor’s comment, in green in the manuscript with track change.

We respond each one of the comments made by the referee during his first revision (yellow) and his second revision (blue) in the manuscript with track change.

We added an acknowledgment to the reviewer, Dr David Crews to express special thanks for the critics and work he did to help us to improve our work.

Considering the referee comments about the poor quality of the English in the text, and we apologize about that, we contracted a native English speaker for the entire revision of the manuscript: Dr. Julian Flavell (http://www.linkedin.com/pub/dir/Julian/Flavell). His corrections are noticed in grey in the manuscript with track change.

A) Editor comments (changes in green in the document)

1) We revised Plos One style requirements.

2) To answer the editor comments about ethical statements (points 2, 3, 4, 5, 6), in the ethical declaration we added the following paragraph:

“All the animals come from private loans of the city of Chetumal (Quintana Roo – Mexico). Verbal consent was requested from the owners, and it was explained how the turtles would be used and for what objective. The individuals on loan were destined exclusively for specific research purposes, respecting all known needs for animal welfare, and were returned to their owners at the end of the data collection.

We were in possession of the permit N ° SGPA / DGVS / 002491/18 issued by Secretaria de Medio Ambiente y Recursos Naturales (SEMARNAT) for the collection of this species. Experimental protocol was approved by Ethical Committee from the “El Colegio de la Frontera Sur”, Mexico.”

3) Page 18 line 334: in Materials and Methods we corrected the mistake in the table caption, it was a typing error.

4) Pages 30-32 lines 618-658: We added the captions for supporting information files.

5) We revised all the references and updated all the new citations.

B) First Revision (in yellow in the document):

Short Title

6) Page 1 Line 10: we deleted the short tittle after we revised Plos One style requirements (see Editor comments).

Summary

7) Page 2 Lines 30-32: According to the reviewer comment, we changed the phrase “Finally, we observed two males in presence of one female and noticed that bolder individuals then displayed courtship behaviors while the shier just ignore the female.” with “Finally, in the simulations that compared two males in the presence of a female we noticed that bolder individuals showed courtship behaviors while the more shy ones simply ignored the female.” The change was made because the result does not come from a single anecdote as the previous sentence suggested, but from multiple individuals and repetitions. 

8) Page 2 Lines 35-36: following the previous correction this sentence should now coincide with results from multiple experiments and not just one, as it previously seemed.

9) Page 2 line 48: we deleted the Key Word after we revised Plos One style requirements (see Editor comments).

Introduction

10) Page 3 line 43-44: as suggested by the reviewer we suppressed the term personality considered as anthropomorphic and controversial. Also, as suggested, we mentioned now the term “reactivity” and suppressed “permanently”.

11) Page 3 line 55: we changed “breed” with “breeding” as suggested by the reviewer.

12) Page 3 line 57: as suggested by the reviewer we added the citation “Gowaty PA, Drickamer LC, Schmid-Holmes S. Male house mice produce fewer offspring with lower viability and poorer performance when mated with females they do not prefer. Anim. Behav. 2003;65: 95-103”.

13) Page 3 line 60: as suggested by the reviewer we added the citation “Ah‐King, M., & Gowaty, P. A. (2016). A conceptual review of mate choice: stochastic demography, within‐sex phenotypic plasticity, and individual flexibility. Ecology and evolution, 6(14), 4607-4642”.

14) Page 4 line 72: as suggested by the reviewer, we added “giving primary importance to survival over reproductive productivity” to specify behaviors.

15) Page 4-5 lines 88-93: as suggested by the reviewer we suppressed the unnecessary sentence and add references (Beach FA and Leboeuf BJ 1967, Coital behaviour in dogs. I. Preferential mating in the bitch. Animal behaviour, 15(4) 546-548, Sánchez-Macouzet O, Rodríguez C, Drummond H. 2014 Better stay together: pair bond duration increases individual fitness independent of age-related variation. Proc. R. Soc. B 281: 20132843).

16) Page 4-5 lines 88-93: As suggested by the reviewer in the discussion, we added the following paragraph in the introduction “There are, for example, cases where behavioral similarity is the key to reproductive success, as in the case of zebra finches (Taeniopygia guttata) and cockatiels (Nymphicus hollandicus) [31-32] ; while, in other species, different or even opposite temperaments can reach high reproductive performances as in the giant panda (Ailuropoda melanoleuca) and in the black rhino (Diceros bocornis) [12, 25, 33].”

17) Page 5 line 94: we created a new sentence as suggested by the reviewer deleting “and the study shows that”

18) Page 5 line 101: as suggested by the reviewer we changed “boldness” for “reactivity”.

19) Page 5 line 108: as suggested by the reviewer we changed “boldness” for “reactivity”.

20) Pace 5 line 113: we changed “attempt to create” with “creating” as suggested by the reviewer. Edit: this correction was subsequently revised and changed by the translator (see correction in gray)

Material and Methods

21) Page 7 line 140: we deleted the hyphenation in “North-western”.

22) Page 7 line 161: we change the tense of “avoid” to “avoided”. Edit: this correction was subsequently revised and changed by the translator (see correction in gray)

23) Page 7 line 164: we were talking about sperm storage that was studied in different turtles’ species but there aren’t any available work about this species. For a better understanding we changed the word “recollecting” with “storing”. Edit: this correction was subsequently revised and changed by the translator (see correction in gray)

24) Page 8 line 167: during the trial we used water only to spraying the individuals, no pool or drinking point were used, therefore there was no reason to refill something. The water we used to spray was always new and after every trial we changed the position of the observation area into a dry one.

25) Page 11 line 249: we changed the tense from “will not be” to “were”.

26) Page 10 line 232: We used “sequentially” to explain that the experiments were done one after the other, but the term is maybe not adequate. We deleted this word to avoid confusion.

Results

27) Page 12 line 271: we changed the word “organisms” with "turtles” as suggested by the reviewer.

28) Page 12 line 274: we changed “didn’t” with “did not” as suggested by the reviewer.

29) Page 12 line 282: as suggested by the reviewer we change “for the shier” by “characteristic of the shier turtles”.

30) Page 12 line 285: As suggested by the reviewer we changed “for the boldest” by “. Variables for the boldest turtles were”.

Discussion

31) Page 20 line 371: we changed the beginning of the sentence from “Also, our” to “Our” as suggested by the reviewer.

32) Page 21 line 398: we deleted “the”.

33) Page 21 line 399: we added “animals”.

34) Page 21 line 407: according to referee’s commentary, we add in the sentence that it was observed for other species “In other species, male competition could be very costly”.

35) Page 22 lines 422-423: in order to rewrite the sentence as suggested by the reviewer, we changed “respond” with “adapt” and we delete “pass under the clutches of” to add “survive to”.

36) Page 22 line 437: according to the reviewer comment we suppressed ”behavioral trait” and added “temperament” to be more specific in our affirmation.

37) Page 22: we deleted the paragraph “In the study of Spoon et al (2006), it can be seen that compatible pairs (male-female) of the Cockatiel Nymphicus hollandicus showed higher levels of coordination in parental care and raised more chicks; another example is given by Carlstead et al (1999a, b) who observed in Black Rhino Diceros bocornis that smaller enclosures determined the breeding of dominant males, while domination decreased in males housed in larger enclosures. In addition, dominance levels were related to reproductive success, with the conclusion that the number of births per year was higher in pairs of rhinos composed of a submissive male and a dominant or aggressive female. This information gives an idea of the optimal conditions for the breeding of captive rhinos, including large enclosures, which facilitate submissive behavior in males, which in turn affects the compatibility of couples; in the giant panda Ailuropoda melanoleuca, Martin-Wintle et al (2017) conducted behavioral studies to further improve reproductive performance and found that specific combinations of temperament traits showed better reproductive performance than others. Males who were more aggressive than their partners were more likely to mate and produce puppies than when the female had a higher level of aggression than the male or even couples with less fearful males, regardless of the fear of the female, obtained better reproductive results. The application of these results to captive breeding management strategies should result in higher reproductive rates and the production of more candidates for the giant panda reintroduction program.” and added a resumed version in page 4 lines 95-100.

38) Page 23 lines 445: we deleted “focused on boldness degree of males”, same with “therefore they could have a” and added “with perhaps”.

39) Page 23 lines 452 and 454: we deleted “has allowed to understand” and added “demonstrates” and added “animals”.

40) Page 23: as suggested by the reviewer and to avoid repetition of the same idea, we deleted the sentence “By another way in term of evolution, being shiest may have some advantages and, if not for reproduction, they may be for antipredatory behaviour, making shiest individuals, that are more prudent in term of exploration, and as we observed in the new environment, generally less predated than boldest individuals as discussed for other species.”

41) Page 24: as suggested by the reviewer we deleted the last paragraph.

Bibliography

42) Following the suggestions of the reviewer we made changes in the citations. We added the following works:

- Gowaty PA, Drickamer LC, Schmid-Holmes S. Male house mice produce fewer offspring with lower viability and poorer performance when mated with females they do not prefer. Anim. Behav. 2003;65: 95-103.

- Ah‐King M, Gowaty PA. A conceptual review of mate choice: stochastic demography, within‐sex phenotypic plasticity, and individual flexibility. Ecol. Evol. 2016;6: 4607-4642.

- Beach FA and Leboeuf BJ Coital behaviour in dogs. I. Preferential mating in the bitch. Anim Behav. 1967; 15: 546-548.

- Sánchez-Macouzet O, Rodríguez C, Drummond H. Better stay together: pair bond duration increases individual fitness independent of age-related variation. Proc. R. Soc. B. 2014; 281: 20132843.

And deleted:

- Cable RN. Oh, Behave: When wildlife behavior matters in Conservation. Mich. J. Sustain. 2013;1

- Cooper Jr WE. Variation in escape behavior among individuals of the striped plateau lizard Sceloporus virgatus may reflect differences in boldness. J. Herpetol. 2009;43:495-502. 

C) Second Revision (in light blue in the document):

Abstract

1) Page 2 line 16: we changed “Temperament variability in animals have consequences for evolution and ecology” to “The variation in temperament among has animals consequences for evolution and ecology”

2) Page 2 line 18: we deleted “works”

3) Page 2 line 18: we deleted “for its measurement”

4) Page 2 line 19: we changed “boldness degree, in the turtle Rhinoclemmys areolata” to “the degree of boldness among individuals Rhinoclemmys areolata”

5) Page 2 line 20: we changed “Secondly, we wanted” to “We also sought”. Edit: this correction was subsequently revised and changed by the translator (see correction in gray)

6) Page 2 line 22: we changed “copula” to “copulation”

7) Page 2 line 22: we changed “male and female” to “males and females”

8) Page 2 line 25: we changed “allowed us to determine” to “distinguish between”

9) Page 2 line 29-30: we changed “rare copula attempts” to “copulation attempts were rarely observed”

Introduction

10) Page 3 line 46: we added the sentence “varying in correlation with the physiology of the animal leading to mediate the response of an individual in different ecological situations” we integrate these aspects following the comment of the reviewer about the importance of the context in behavioral interactions.

11) Page 3 line 46: we added the citations “Clarke et al 1995” and “Michelangeli et al 2017”

12) Page 3 line 47: we added the citation “Michelangeli et al 2019”

13) Page 4 line 68-75: we rewrite the paragraph in response to what behaviors we were considering “In general, bolder individuals tend to be more active, feed more often and in risky areas as they are more likely to explore and get away from safe and well-known locations [22]. This temperament, however, leads to an increased risk of encountering predators [23] and exposing themselves to parasites [24], decreasing their survival rate. On the other hand, shy individuals engage in opposite strategy giving primary importance to survival over reproductive productivity. This implies that this temperament would allow to regulate survival with other needs of the species [21].”

14) Page 4 line 71: we added the citation “Wilson et al 1994”

15) Page 4 line 72: we added the citation “Sasaki et al 2018”

16) Page 4 line 73: we added the citation “Gharnit et al 20202

Material and Methods

17) Pages 7-8 lines 162-165: we added a paragraph “Female turtles are able to retain vital sperm for up to 4 years, but it is not yet clear in which species therefore we cannot exclude that it can also occur in R. areolata [51].” to better explain the capacity of female turtles to storage sperm. Edit: this correction was subsequently revised and changed by the translator (see correction in gray)

18) Page 8 line 169 and 177: we explained in two parts “The water in the maintenance area was changed daily.” and “There were no food or small pools in the experimental area in order not to affect the behaviors [41].”, respectively, that the water in the maintenance area was changed every day, while in the experimental area there was never any use of pools with water, therefore there was no presence of pheromones of other individuals.

19) Page 11 line 249: we changed “not used more than once” to “only used once”

Results

20) Page 12 line 267: we changed “We observed 17 behavioral traits (S1 Table) but several of them were displayed by all the turtles and were not useful to distinguish temperaments; therefore, we selected 12 behavioral traits that were used as variables” to “Of 17 behavioral traits observed (S1 Table), we selected 12 for study”

Discussion

21) Page 20 line 381: we changed “the” to “he”

22) Page 20 line 385: we changed “copula” to “copulation”

23) Page 20 line 385: we deleted “,and in” and started a new sentence “In the common”

24) Page 20 lines 385-391: we modified the paragraph “where males prefer the more active males when they find themselves alone and without the presence of predators in their surroundings, whereas when shier specimens find themselves alone in the presence of a female, they spend time and energy in courtship behaviour [47].” to “males’ behavior together with the predatory context influence the mating preference of the females. Those, which have not been exposed to predatory signals prior to mating, mate more often with more active males. If, on the other hand, these have been subjected to predatory signals, they tend to choose less active males with likely a longer life expectancy and to avoid the more active males in order to increase the survival of the offspring in an environment with risk of predation [53].”

25) Page 21 lines 393-394: we modified the sentence “boldest individuals court the female only in presence of a potential competitor and directly mate with” to “the boldest individuals court females, but only if another male is present; otherwise these males immediately mount and attempt to copulate.”

26) Page 21 line 395: we modified “courtship the female only if no competitor is present and is not very effective at mating.” to “males will only court if other males are absent; even then mating is rare.”

27) Page 21 line 398: we deleted “for”

28) Page 21 line 398: we changed “that more time is necessary to display a mating behavior” to “shier animals require more time to display courtship and mating behavior”

29) Page 21 line 399: we deleted the sentence “anyway our work shows apparent differences in the sexual behavior according to males’ temperaments.”

30) Page 21 line 402: we changed “specimens” to “individuals”

31) Page 21 line 403: we modified the sentence “On the other hand, it avoids competition and the possibility of losing energy when a bolder male is competing.” to “Shier turtles could avoid injuries and the possibility of losing energy when a bolder male is competing. In our experimental contexts, we could not directly observe injuries phenomena between males, however we did observed, but not during those experiment, that males have bitten each other trying to approach the same female.” to explain that bolder individuals can hurt opponents if they are led to do so.

32) Page 21 line 408: we deleted “Therefore we cannot exclude that the temperament may have an influence on competition” since we explained the phenomena in the prior paragraph.

33) Page 22 line 433: we changed the sentence “To our best knowledge, this is the first work that connects behavioral trait to reproductive behaviors in land turtles” to “Assumptions have been made in the past about the likely relationship between temperament and reproductive behaviors, but to our best knowledge, this is the first work that formalizes it experimentally for land turtles” given that as the reviewer commented, Auffenberg talked about the possibility of the existence of shy specimens during mating in the genus Gopherus, but until now it had not been formally tested and proven for any land turtle species. We also added the reference “Auffenberg 1996”.

34) Page 23 line 452: we changed “specimens” to “individuals”.

Acknowledgments

35) Page 23 line 462: we wanted to add an acknowledgment to the reviewer “We would like to express special thanks to Dr David Crews, who has critically read our manuscript, providing useful and substantial comments that have allowed us to improve our work.”

Bibliography

36) Following the suggestions of the reviewer we made changes in the citations. We added the following works:

- Clarke AS, Boinski S. Temperament in nonhuman primates." American Journal of Primatology. Am. J. Primatol. 1995;37: 103-125.

- Michelangeli M, Goulet CT, Kang HS, Wong BB, Chapple DG. Integrating thermal physiology within a syndrome: Locomotion, personality and habitat selection in an ectotherm. Funct. Ecol. 2018;32: 970-981.

- Michelangeli M, Chapple DG, Goulet CT, Bertram MG, Wong BB. Behavioral syndromes vary among geographically distinct populations in a reptile. Behav. Ecol. 2019;30: 393-401.

- Wilson DS, Clark AB, Coleman K, Dearstyne T. Shyness and boldness in humans and other animals. Trends Ecol. Evol. 1994;9: 442-446.

- Sasaki, T., Mann, R. P., Warren, K. N., Herbert, T., Wilson, T., & Biro, D. Personality and the collective: bold homing pigeons occupy higher leadership ranks in flocks. Philos. Trans. R. Soc. Lond., B, Biol. Sci. 2018;373:20170038.

- Gharnit E, Bergeron P, Garant D, Réale D. Exploration profiles drive activity patterns and temporal niche specialization in a wild rodent. Behav. Ecol. 2020.

- Auffenberg W. On the courtship of Gopherus Polyphemus. Herpetologica, 1966;22: 113-117.

D) Translator comments (changes in grey in the document)

Abstract

1) Page 2 line 17: we deleted “points to” and substitute it with “is on”

2) Page 2 line 28: we added “authors”

3) Page 2 line 19: we changed “we first wanted to” with “our first aim was”

4) Page 2 line 20: we changed “we also sought to” with “our second aim was to”

5) Page 2 line 22: we deleted “the”

6) Page 2 line 23: we changed “We first observed the males in four different situations and recorded 17 behavioral traits.” with “Males were observed in four different situations and 17 behavioral traits were recorded.” 

7) Page 2 line 25: we deleted “to” and “best”

8) Page 2 line 25: “we” and added “and”

9) Page 2 line 26: we deleted “the” and added “individuals”

10) Page 2 line 29: we changed “attempt” to “attempted”

11) Page 2 line 31: we changed “showed” to “displayed”

12) Page 2 line 32: we changed “the more shy” to “the shier”

13) Page 2 line 33: we deleted “the”

14) Page 2 line 34: we changed “to modulate” to “in modulating”

15) Page 2 line 36: we changed “try to copulate only” to “only try to copulate”

16) Page 2 line 37: we changed “those” to “these”

Introduction

17) Page 3 line 43: we changed “what some scientists call” to “termed”

18) Page 3 line 44: we added “by scientists”

19) Page 3 line 51: we changed “great influence” to “significant”

20) Page 3 line 52: we changed “exhibits” with “exhibitions”

21) Page 3 line 55: we deleted “in”

22) Page 3 line 55: we changed “success of coupling” to “coupling success”

23) Page 4 line 65: we added a comma 

24) Page 4 line 67: we changed “seems” to “appears”

25) Page 4 line 69: we changed “get away” to “move away”

26) Page 4 line 71: we added “an” before opposite

27) Page 4 line 74: we changed “would allow to regulate survival with” to “tries to achieve a balance between survival and”

28) Page 4 line 77: we changed “Knowing these” to “Knowledge on these”

29) Page 4 line 81: we deleted “,in fact,”

30) Page 4 line 86: we added an hyphen

31) Page 5 line 91: we changed “reach” to “attain”

32) Page 5 line 99: we changed “, as it could help stabilize social systems and reduce” to “contributing to the stabilization of social systems and reducing”

33) Page 5 line 102: we changed “straighten up” to “become upright after”

34) Page 5 line 102: we changed “his” to “its”

35) Page 5 line 104: we changed “to insert” to “in placing”

36) Page 5 line 108: we deleted “to” before “reactivity”

37) Page 5 line 108: we changed “to measure” to “while researching”

38) Page 5 line 109: we added “the” before “Agassiz’s”

39) Page 5 line 109: we deleted “the” and capitalized “Boldness”

40) Page 5 line 110: we changed “using the time” to “by recording the time taken”

41) Page 5 line 111: we deleted “the time”

42) Page 5 line 111: we changed “Therefore” to “Evidently”

43) Page 5 line 112: we changed “different” to “diverse”

44) Page 5 line 112: we deleted “but” and added “; however,”

45) Page 5 line 113: we substituted “of them” with “method”

46) Page 5 line 113: we substituted “For this reason” with “Therefore”

47) Page 6 line 116: we changed “a lack of work addressing specifically” to “very little research that specifically addresses”

48) Page 6 line 117: we changed “if it influences” to “its influence on”

49) Page 6 line 119: we changed “determinate” to “determine the”

50) Page 6 line 119: we substitute “to determine the temperament in these species that can also be used for others terrestrial turtle species” with “that not only ascertains temperament for this species but also other species of terrestrial turtle”

51) Page 6 line 121: we changed “that may modulate” to “in modulating”

52) Page 6 line 122: we deleted “the” before “competition”

Materials and methods

53) Page 6 line 127: we substituted “of” with “from”

54) Page 6 line 141: we changed “It is used as pet, food and in the traditional medicine” to “It is used as a pet, for food and traditional medicine”

55) Page 7 line 145: we changed “has” with “presents a”

56) Page 7 line 147: we changed “but” to “; however,”

57) Page 7 line 147: we deleted “that” before “is not”

58) Page 7 line 148: we changed “; however, these derive” to “, although this information solely derives”

59) Page 7 line 149: we changed “them” to “the individuals”

60) Page 7 line 153: we changed “It can be seen the difference between the tails (red arrow), “ to “Differences between the tails (red arrow) are visible;”

61) Page 7 line 161: we substituted “avoided” with “prevented”

62) Pages 7-8 lines 162-165: we substituted “It also avoids female from storing semen from males and be less reactive to males. Female turtles are able to retain vital sperm for up to 4 years, but it is not yet clear in which species therefore we cannot exclude that it can also occur in R. areolata [51].” with “It also ensured that the female did not store semen and therefore be less reactive to males. Females from some species of turtle are able to retain vital sperm for up to 4 years; however, it is not yet clear in which species, therefore we cannot dismiss that it could also occur in R. areolata [51].”

63) Page 8 line 166: we substituted “was” with “resembled”

64) Page 8 line 166: we deleted “to” before “the natural”

65) Page 8 line 167: we deleted “it was possible”

66) Page 8 line 171: we changed “keep the best” to “sustain optimum”

67) Page 8 line 177: we substituted “in case details of the events were needed” with “in the event that details of specific events were required”

68) Page 8 line 178: we substituted “in order not to affect the behaviors” with “that could have had an effect on turtle behavior”

69) Page 8 lines 181-183: we changed “Each simulation took place in open air, on sunny days, during hours of maximum activity of turtles (8.30am-11.30am and 05.30pm - 06.30pm), when the temperatures are not very high” to “Each simulation took place in the open air, on sunny days and during the hours of maximum turtle activity (8.30am-11.30am and 05.30pm - 06.30pm) when temperatures were not very high”

70) Page 8 lines 185-188: we changed “The observation area was moved in an adjacent area after every simulation to reduce waiting time between experiments and recovery time of environmental conditions, then a new group of turtles was placed to repeat simulation” to “The observation area was moved to an adjacent area after each simulation to reduce the time elapsed between experiments and the recovery time for environmental conditions; subsequently a new group of turtles was placed into the observation area in order to repeat the simulation”

71) Page 9 line 191: we changed figure 4 tagline to “Observation area viewed from the side and above”

72) Page 9 line 196: we changed “reason” to “aim”

73) Page 9 line 197: we changed “taken” to “removed” and we deleted “the” before “confinement”

74) Page 9 line 199: we changed “taken” to “removed”

75) Page 9 lines 201-202: we changed “in” to “into the”

76) Page 9 line 203: we changed “taken” to “removed”

77) Page 9 line 202: we changed “in” to “into the”

78) Page 9 line 202: we substituted “released” with presented

79) Page 9 line 206: we changed “To organize males in order to describe their boldness-shyness level” to “In order to organize males according to their level of boldness-shyness”

80) Page 9 lines 210-211: we changed “it was measured the moment of abrupt change of slope within data range, this way it was possible to classify male” to “the moment an abrupt change of slope occurred within the data range was recorded, allowing the classification of males”

81) Page 10 line 216: we changed “Afterwards” to “Subsequently”

82) Page 10 line 217: we changed “allowed us to elaborate” to “enabled the creation of”

83) Page 10 line 219: we changed “variables were analysed through a Mann-Whitney test to determine which ones” to “a Mann-Whitney test was applied to determine which variables were significant”

84) Page 10 line 222: we substituted “starting” with “initiating”

85) Page 10 line 223: we changed “linked with” to “a good indication of”

86) Page 10 lines 223-225: we substituted “We did this test to ensure results are not the consequence of the size or age of the turtles” with “We implemented this test to ensure that our results were not the consequence of turtle size or age”

87) Page 10 line 225: we substituted “The two” with “Both”

88) Page 10 line 226: we substituted “through” with “by applying a”

89) Page 11 lines 239-240: we substituted ” it was measured the moment of abrupt change of slope within data range” with “the moment an abrupt change of slope occurred within the data range was recorded”

90) Page 11 line 243: we changed “that allowed us to elaborate” to “allowing the creation of”

91) Page 11 line 251: we substituted “freed” with “released”

92) Page 11 lines 257-258: we changed “it was measured the moment of abrupt change of slope within data range” to “the moment an abrupt change of slope occurred within the data range was recorded”

93) Page 11 line 261: we changed “us to elaborate” to “that allowed the creation of”

Results

94) Page 12 line 268: we substituted “each of them” with “these traits”

95) Page 12 line 270: we deleted “as shier”

96) Page 12 line 271: we changed “that had an” to “with an”

97) Page 12 line 272: we added “as shier”

98) Page 12 line 272: we changed “bolder were those that had an index greater than 0.5.” to “those with an index greater than 0.5 bolder.”

99) Page 12 line 273: we changed “This allowed us to divide individuals” to “As a result, individuals were divided”

100) Page 12 line 283: we changed “high” to “a high duration of”

101) Page 13 Table 1: we changed “peluche” with “stuffed toy”

102) Page 15 line 302: we added “only” before “used”

103) Page 15 line 303: we changed “the ones” to “those”

104) Page 15 line 304: we changed “show” with “present any”

105) Page 15 line 305: we substituted “We defined if they” with “We determined if the behaviors”

106) Page 15 line 306: we changed “are linked with” to “corresponded to”

107) Page 15 line 309: we changed “they were” to “this was”

108) Page 15 line 310: we changed “are linked with” to “displayed the following behaviours”

109) Page 15 lines 314-315: we changed “None of those behaviors is displayed by shiest individuals, while the boldest are more linked with” to “None of these behaviors was displayed by the shiest individuals, while the boldest were more associated with”

110) Page 16 line 319: we revised the table tagline changing “must-dimensional” with “multi-dimensional”

111) Page 17 line 323: we revised the table tagline changing “must-dimensional” with “multi-dimensional”

112) Page 17 line 328: we added “the” before “temperament”

113) Page 17 line 331: we substituted “On the contrary” with “In contrast”

114) Page 17 line 331: we added “individuals” after “shiest”

115) Page 17 line 331: we changed “except for” to “with the exception of”

116) Page 18 line 333: we revised the table tagline changing “must-dimensional” with “multi-dimensional”

117) Page 18 line 337: we revised the description of Figure 5 from “The red circle shows variables linked with the boldest individuals, the blue circle identifies behaviors linked to the shiest ones.” to “The red circle shows variables associated with the boldest individuals; the blue circle identifies behaviors associated with the shiest ones.”

118) Page 19 line 347: we revised the description of Figure 7 from “The red curve includes variables linked with boldest individuals, the blue curve congregates behaviors linked with the shiest individuals.” to “The red curve includes variables associated with the boldest individuals; the blue curve congregates behaviors associated with the shiest individuals.”

119) Page 19 line 350: we revised the description of Figure 8 from “The red curve encompasses variables linked with boldest individuals, the blue curve includes behaviors linked with the shiest ones.” to “The red curve encompasses variables associated with the boldest individuals; the blue curve includes behaviors associated with the shiest individuals.”

120) Page 19 line 353: we revised the description of Figure 9 from “The red circle delimits variables highly linked with boldest males in competition with each other, the yellow circle encompasses those highly linked with a bold male in competition with a shy one, the green ellipse includes behaviors linked with the boldest not considering who is its competitor.” to “The red circle delimits variables that are highly associated with the boldest competing males; the yellow circle encompasses those highly associated with a bold male competing with a shy individual; the green ellipse includes behaviors associated with the boldest individual regardless of competitor.”

Discussion

121) Page 19 line 360: we deleted “but” before “separately”

122) Page 19 line 361: we changed “them” with “these methods”

123) Page 19 line 363: we substituted “purpose of looking for” with “aim of finding”

124) Page 19 line 363: we changed “the most accurate” to “accurately”

125) Page 19 line 365: we changed “seem to be the best to determine the” to “appear to be the most reliable for determining”

126) Page 19 line 366: we changed “being quiet (shiest) or not” to “quiet (shiest) or not quiet”

127) Page 20 line 367: we substituted “a lot” with “frequently”

128) Page 20 line 368: we deleted “of” before “previous”

129) Page 20 line 369: we changed “shows” to “demonstrates”

130) Page 20 line 369: we added “taken” after “time”

131) Page 20 line 370: we changed “linked with the” to “associated with”

132) Page 20 line 370: we changed “the determination” to “determining”

133) Page 20 line 372: we substituted “if” with “though”

134) Page 20 line 372: we substituted “equivalent whatever” with “comparable irrespective”

135) Page 20 line 373: we changed “strategy of life” to “life strategy”

136) Page 20 line 374: we changed “linked” to “associated”

137) Page 20 line 375: we changed “be a better method, we suggest further studies to test it also” to “provide an improved method, we suggest further studies that examine this”

138) Page 20 line 384: we substituted “looking” with “searching”

139) Page 20 line 384: we substituted “having” with “presenting”

140) Page 20 line 387: we changed “,which” to “that”

141) Page 20 line 389: we changed “with likely” to “that are more likely to have”

142) Page 21 line 392: we added “turtle” after “shier”

143) Page 21 line 396: we changed “towards” to “in the presence of”

144) Page 21 line 405: we changed “injuries” to injury”

145) Page 21 line 406: we changed “we did observed, but not during those experiment, that males have bitten each other” to “, in other experiments, we have observed that males have bitten each other when”

146) Page 21 line 408: we changed “and refraining those costs could also be” to “Some reptiles avoid or reduce their response to such competition and could provide”

147) Page 21 line 410: we substituted “avoid” with “prevent”

148) Page 21 line 410: we added “costs” after “predators”

149) Page 21 line 412: we changed “intensity and costs of the fights” to “fight intensity and resulting costs”

150) Page 21 line 413: we changed “dependant of the temperament is under environmental constraint” to “dependent on temperament is to some extent subject to environmental constraints”

151) Page 22 line 418: we substituted “Moreover” with “Furthermore”

152) Page 22 lines 418-420: we changed “by isolated males when a predator is close to the female so they were able to balance risk of predation with courtship behavior” to “displayed by isolated males when a predator is close to the female, allowing them to balance predation risk with courtship behavior”

153) Page 22 line 425: we substituted “those” with “these”

154) Page 22 line 436: we changed “What we have achieved” to “The achievements in this study”

155) Page 22 line 437: we changed “might compose a” to “may contribute to”

156) Page 23 line 447: we substituted “to keep” with “of conserving”

157) Page 23 line 449: we deleted “the”

158) Page 23 line 452: we substituted “likely” with “probably”

159) Page 23 line 559: we deleted “needed”

Supporting information

160) Page 30 line 621: we changed the description of Table S1 from “In the first column, there are names of each individual, in grey background color the bolder and in white the shier.” to “The first column shows the names of each individual, bolder individuals have a grey backgrounds while those considered shier have white.”

161) Page 30 line 625: we changed the description of Table S2 from “In grey background color the individuals considered bolder, in white those considered shier.” to “Individuals considered bolder have grey backgrounds while those considered shier have white.”

162) Page 30 line 630: we changed the description of Table S3 from “In the first column, there are names of each individual, in grey background color the bolder and in white the shier.” to “The first column shows the names of each individual. Bolder individuals have grey backgrounds while shier individuals have white.”

163) Page 30 line 635: we changed the description of Table S4 from “In the first column, there are names of each individual, in grey background color the bolder and in white the shier.” to “The first column shows the names of each individual, bolder individuals have grey backgrounds while shier have white.”

164) Page 30 line 640: we changed the description of Table S5 from “In the first column, there are names of each individual, in red the bolder and in blue the shier. Columns 2,3,4 and 5 indicate pairs by temperament (BB=boldVSbold, BS=boldVSshy, SS=shyVSshy, SB=shyVSbold).” to ” The first column shows the names of each individual, bolder individuals have red backgrounds while shier have blue. Columns 2,3,4 and 5 indicate pairs by temperament (BB=boldVSbold, BS=boldVSshy, SS=shyVSshy, SB=shyVSbold).”

165) Page 30 line 657: we changed the description of S2 Video from “Shy is walking around the arena while the bold courtships the female.” to “Shy male is walking around the arena while bold male courtships the female.”

---

## [Decision Letter · Decision Letter 2]

14 Dec 2020

Temperament and sexual behaviour in the Furrowed Wood Turtle Rhinoclemmys areolata

PONE-D-20-14068R2

Dear Dr. Henaut,

We’re pleased to inform you that your manuscript has been judged scientifically suitable for publication and will be formally accepted for publication once it meets all outstanding technical requirements.

Kind regards,

Cheryl S. Rosenfeld, DVM, PhD

Section Editor

PLOS ONE

Additional Editor Comments (optional):

Reviewers' comments:

Reviewer's Responses to Questions

**Comments to the Author**

1. If the authors have adequately addressed your comments raised in a previous round of review and you feel that this manuscript is now acceptable for publication, you may indicate that here to bypass the “Comments to the Author” section, enter your conflict of interest statement in the “Confidential to Editor” section, and submit your "Accept" recommendation.

Reviewer #1: All comments have been addressed

2. Is the manuscript technically sound, and do the data support the conclusions?

Reviewer #1: Yes

3. Has the statistical analysis been performed appropriately and rigorously? 

Reviewer #1: Yes

4. Have the authors made all data underlying the findings in their manuscript fully available?

Reviewer #1: Yes

5. Is the manuscript presented in an intelligible fashion and written in standard English?

Reviewer #1: Yes

6. Review Comments to the Author

Reviewer #1: Overall, excellent responses. Here are some more minor changes that would aid in introduction.

LL 25-26: Change to “…shier individuals; five behavioral …” OR “…shier individuals. Five behavioral…”

LL 46-47: Change to “…animal [6-7] mediating the response…”

LL 48: Change to “Behavioral differences in animals…”

LL 55: Change to “…(copulation success)…”

LL 61: Word choice. Perhaps “consistently” rather than “systematically”

7. PLOS authors have the option to publish the peer review history of their article (what does this mean?). If published, this will include your full peer review and any attached files.

Reviewer #1: **Yes: **David Crews

---

## [Editor Report · Acceptance letter]

16 Dec 2020

PONE-D-20-14068R2 

Temperament and sexual behaviour in the Furrowed Wood Turtle *Rhinoclemmys areolata*

Dear Dr. Henaut:

I'm pleased to inform you that your manuscript has been deemed suitable for publication in PLOS ONE. Congratulations! Your manuscript is now with our production department. 

Kind regards, 

on behalf of

Dr. Cheryl S. Rosenfeld 

Section Editor

PLOS ONE